# Beyond Amyloid and Tau: The Critical Role of Microglia in Alzheimer’s Disease Therapeutics

**DOI:** 10.3390/biomedicines13020279

**Published:** 2025-01-23

**Authors:** Daniela Dias, Renato Socodato

**Affiliations:** 1i3S—Instituto de Investigação e Inovação em Saúde, Universidade do Porto, 4099-022 Porto, Portugal; daniela.dias@i3s.up.pt; 2ESS—Escola Superior de Saúde do Politécnico do Porto, 4200-072 Porto, Portugal; 3IBMC—Instituto de Biologia Molecular e Celular, 4200-135 Porto, Portugal

**Keywords:** Alzheimer’s disease, microglia, amyloid-beta, tau pathology, neuroinflammation, neuron–microglia crosstalk, neurodegeneration, therapeutic strategies

## Abstract

Alzheimer’s disease (AD) is traditionally viewed through the lens of the amyloid cascade hypothesis, implicating amyloid-beta and tau protein aggregates as the main pathological culprits. However, burgeoning research points to the brain’s resident immune cells, microglia, as critical players in AD pathogenesis, progression, and potential therapeutic interventions. This review examines the dynamic roles of microglia within the intricate framework of AD. We detail the involvement of these immune cells in neuroinflammation, explaining how their activation and response fluctuations may influence the disease trajectory. We further elucidate the complex relationship between microglia and amyloid-beta pathology. This study highlights the dual nature of these cells, which contribute to both aggregation and clearance of the amyloid-beta protein. Moreover, an in-depth analysis of the interplay between microglia and tau unveils the significant, yet often overlooked, impact of this interaction on neurodegeneration in AD. Shifting from the conventional therapeutic approaches, we assess the current AD treatments primarily targeting amyloid and tau and introduce novel strategies that involve manipulating microglial functions. These innovative methods herald a potential paradigm shift in the management of AD. Finally, we explore the burgeoning field of precision diagnosis and the pursuit of robust AD biomarkers. We underline how a more profound comprehension of microglial biology could enrich these essential areas, potentially paving the way for more accurate diagnostic tools and tailored treatment strategies. In conclusion, this review expands on the conventional perspective of AD pathology and treatment, drawing attention to the multifaceted roles of microglia. As we continue to enhance our understanding of these cells, microglial-focused therapeutic interventions emerge as a promising frontier to bolster our arsenal to fight against AD.

## 1. Introduction

Alzheimer’s disease (AD) is a progressive, debilitating neurodegenerative disorder that predominantly affects individuals in their mid-60s and older [1]. AD is characterized by an insidious onset, gradual progression, and a decline in multiple cognitive domains, significantly affecting memory, executive function, and language skills [2]. AD is the most common cause of dementia, accounting for 60–70% of all dementia cases [1,2]. As the global population ages, the number of individuals affected by AD is projected to rise sharply, emphasizing the urgency for research into understanding its pathology, identifying risk factors, and developing effective treatments [1].

The physical, psychological, and economic burden of AD is immense, not just for the individuals suffering from the disease but also for their families, caregivers, and society [3,4]. The impact on the individual involves progressive loss of independence, changes in personality and behavior, and ultimately, complete dependence on caregivers [3,4]. Caregivers often face significant stress, including emotional distress, physical exhaustion, and financial strain [3,4]. Society bears the burden of the rising cost of healthcare and long-term care services, loss of productivity, and the challenge of meeting the needs of an aging population [5].

### 1.1. AD Overview, Epidemiology, and Health Economics

AD, first described by German psychiatrist Alois Alzheimer in 1906, is a chronic neurodegenerative disorder characterized by a progressive decline in cognitive abilities and memory [1,2]. Primarily affecting older adults, age is the most significant risk factor for AD [6]. However, it is essential to note that AD is not a normal part of aging. Symptoms typically manifest after the age of 60, but early-onset AD can occur in individuals as young as their 40s [6].

The pathological hallmarks of AD are extracellular plaques composed of amyloid-beta (Aβ) peptides and intracellular neurofibrillary tangles (NFTs) made up of hyperphosphorylated tau proteins [1,2]. Aβ peptides, derived from the larger amyloid precursor protein (APP) through sequential cleavage by beta and gamma-secretases, accumulate and aggregate into insoluble Aβ plaques in AD [1,2]. Tau, a microtubule-associated protein involved in microtubule assembly and stability, undergoes abnormal hyperphosphorylation in AD, detaching from microtubules and aggregating into NFTs [1,2]. These plaques and tangles disrupt neuronal function and lead to cell death [1,2].

The progression of AD is typically divided into preclinical, mild cognitive impairment (MCI), and dementia stages [7]. Each stage is characterized by distinct cognitive and functional changes that progressively worsen over time [7]. The preclinical stage of AD is characterized by the presence of amyloid plaques and NFTs in the brain, even though individuals do not show symptoms [7]. This stage can last for years or even decades before symptoms appear. During this stage, the disease affects the brain and lays the groundwork for future cognitive decline [7].

Noticeable cognitive changes mark the transition to the MCI stage. Individuals at this stage exhibit memory and thinking problems that are noticeable to themselves or others, but these changes are not severe enough to interfere with daily activities [7]. The annual probabilities of transition to more severe states were found to be 8% for normal cognition and 22% for MCI due to AD at age 65, and these rates increase as a function of age [8,9]. Once these impairments progress sufficiently to affect daily functioning, an individual is diagnosed with AD dementia. This stage is characterized by significant memory loss, confusion, difficulty communicating, disorientation, and behavioral changes [8,9]. The progression rates from normal cognition to MCI due to AD range from 4% to 10% annually [8,9]. In conclusion, the progression of AD from the preclinical stage to MCI and then to dementia is a complex process that involves significant changes in cognitive function [8,9]. Understanding these stages and the factors that influence progression can help in the early diagnosis and management of AD [7].

The likelihood of developing AD dementia increases as individuals move from normal cognition without amyloid-beta (Aβ) accumulation to early neurodegeneration and subsequently to mild cognitive impairment (MCI) [7,8,9]. For those with Aβ deposition and neurodegeneration, the projected lifetime risk of AD dementia is 41.9% for women and 33.6% for men [7,8,9]. Various factors have been identified as risk factors for AD, including a low score on the Mini-Mental State Examination (MMSE), a high score on the Alzheimer’s Disease Assessment Scale-cognitive subscale (ADAS-cog), positive APOE4 status, white matter hyperintensity volume, entorhinal cortex atrophy, cerebrospinal fluid (CSF) total tau levels, CSF neurogranin levels, dependency on instrumental activities of daily living (IADL), and being female [7,8,9].

The economic burden of AD is substantial and multifaceted, encompassing direct medical costs, indirect costs such as lost productivity, and the immense emotional and physical toll on caregivers [3,4,5]. Direct medical costs for AD are high and include diagnostic services, hospital care, medical services, home health care, and long-term care facilities [3,4,5]. Indirect costs, such as lost patient and caregiver productivity, also contribute to the economic burden [3,4,5]. Patients with AD often retire early due to their declining cognitive function, and caregivers may reduce their working hours or stop working entirely to provide care [3,4,5].

AD is expected to have a significant economic impact as the world population ages. In the United States, the cost of care for people with AD is projected to reach $1.1 trillion by 2050 [3,4,5], and in Europe, costs are expected to increase due to increased patient numbers and higher costs. These rising expenses will significantly strain healthcare systems, highlighting the need for effective interventions that can delay disease onset, slow progression, and reduce the burden of care [4]. Addressing this growing public health issue will require comprehensive solutions to ensure a better quality of life for those affected by AD and their caregivers [3,4,5].

The profound socio-economic impact of AD, driven by its escalating prevalence and multifaceted pathology, underscores the urgent need for innovative therapeutic interventions. While the amyloid cascade and tau hypotheses have traditionally dominated the research landscape, there is growing recognition of the crucial role of neuroinflammation and microglial dysfunction in driving disease progression. Microglia, the brain’s resident immune cells, emerge as potential culprits and therapeutic targets in this complex interplay of pathological processes.

Microglia play a central role in maintaining cerebral homeostasis by actively engaging in amyloid clearance, synaptic remodeling, and inflammatory regulation. However, their dysregulation can tip the balance toward neurodegeneration and worsen the toxic milieu of AD. This dynamic role increases their relevance to the disease and highlights their potential as key targets for therapeutic intervention. To untangle this complexity, it is essential to investigate the biology and functions of microglia in both health and disease.

### 1.2. Microglia: The Brain’s Immune Sentinels

Microglia, colloquially known as the immune sentinels of the brain, are integral to the active immune defense in the central nervous system (CNS) [10,11,12]. As descendants of yolk-sac-derived myeloid progenitors during embryonic development, microglia are the largest immune cell population in the CNS [10,11,12]. Their reach extends across various brain regions, with a higher concentration in white matter areas, and they comprise approximately 10–15% of all brain cells [10,11,12].

Microglia are incredibly dynamic, continuously shifting and adapting to the changing needs of the CNS [10,11,12,13,14]. They maintain a watchful eye on their surroundings in their basal, or “surveillant,” state. In this state, microglia exhibit a ramified morphology characterized by small cell bodies and intricate, highly branched processes that extend like the fingers of a hand. These processes are perpetually in motion, reaching out and retracting to scan the local microenvironment for abnormalities [10,11,12,13,14].

Microglia are susceptible to environmental changes and can detect subtle shifts indicative of infection, trauma, or other neural distress. Upon identifying such signals, microglia transition from their “surveillant” state to an “activated” state, marked by significant morphological and functional changes [10,11,12,13,14]. The previously delicate branching processes retract, and the cell body enlarges, giving the activated microglia a more amoeboid appearance. This transformation enables them to swiftly navigate the site of injury or infection [10,11,12,13,14].

Activated microglia are not just mobile; they are also functional powerhouses [10,11,12,13,14]. To orchestrate an immune response, they release various inflammatory mediators, including cytokines, chemokines, and reactive oxygen species [10,11,12,13,14]. Their phagocytic activity—the ability to engulf and degrade harmful entities—is greatly enhanced, enabling them to effectively clear cellular debris, infectious agents, and aggregated proteins [10,11,12,13,14].

However, the role of microglia extends far beyond the realms of immune defense. These cells actively shape the intricate neural circuits that form the foundation of cognition and behavior [10,11,12,13,14]. Through a process known as synaptic pruning, microglia selectively eliminate weaker synaptic connections while strengthening others, thereby fine-tuning the synaptic network [10,11,12,13,14]. This process is vital during brain development and essential for adult learning and memory [10,11,12,13,14].

Furthermore, microglia contribute to neuronal survival and health by secreting a range of trophic factors, such as nerve growth factor (NGF), brain-derived neurotrophic factor (BDNF), and glial cell-derived neurotrophic factor (GDNF) [10,11,12,13,14]. These substances support neuronal growth, differentiation, and survival, highlighting the supportive role of microglia in neuronal health [10,11,12,13,14].

Microglia also contribute to learning and memory by influencing synaptic plasticity, the ability of synapses to strengthen or weaken over time [10,11,12,13,14]. They release signaling molecules that can modify the strength of synaptic connections, affecting neuronal communication and, thus, cognitive functions [10,11,12,13,14].

However, when the microglial activity becomes dysregulated or overactivated, the consequences can be dire, particularly in the context of neurodegenerative diseases [10,11,12,13,14]. In conditions like AD, persistent microglial activation can lead to chronic neuroinflammation, contributing to neuronal damage and cognitive decline [10,11,12,13,14]. This underscores the delicate balance that microglia must maintain; while their protective functions are critical for CNS health, unchecked microglial activity can tip the scale toward pathology [10,11,12,13,14]. Therefore, understanding the various facets of microglial biology—from their developmental origins to their diverse functional roles—is essential for developing effective therapeutic strategies for neurodegenerative diseases [10,11,12,13,14].

## 2. Microglial States, Amyloid, and Tau in AD

### 2.1. Microglial Activation in AD: A Spectrum of Functional States

In the unique context of AD, neuroinflammation characterizes the intricate inflammatory responses that persist within the brain. Microglia, the chief immune cells in the brain, are central players in this inflammatory cascade. Neuroinflammation, primarily driven by microglial activation, plays a pivotal role in the pathogenesis of AD, influencing disease progression and severity [15,16].

Traditionally, microglial activation has been categorized into pro-inflammatory and anti-inflammatory states. In the pro-inflammatory state, also referred to as the M1 state, microglia are characterized by their production and release of pro-inflammatory mediators, such as cytokines like interleukin-1β (IL-1β) and tumor necrosis factor-α (TNF-α), chemokines, and reactive oxygen species (Figure 1). This type of activation is generally concurrent with neuronal injury and promotes the progression of neurodegenerative diseases.

The pro-inflammatory state of microglia is characterized by the upregulation of several surface markers, including major histocompatibility complex class II (MHC II), CD68, and CD86 [13]. These microglia produce high levels of pro-inflammatory cytokines, such as IL-1β, IL-6, TNF-α, and chemokines like CCL2 and CXCL10. They also produce reactive oxygen species and nitric oxide, which can cause neuronal damage. The pro-inflammatory state is typically induced by exposure to lipopolysaccharide (LPS), a component of the cell wall of gram-negative bacteria, or by pro-inflammatory cytokines like interferon-γ (IFN-γ) [13].

In contrast, the anti-inflammatory state, also referred to as the M2 state, is thought to be triggered by a specific set of factors, including anti-inflammatory cytokines such as IL-4 and IL-13 (Figure 1). Microglia in this state exhibit anti-inflammatory characteristics and play a role in tissue repair and the resolution of inflammation. This occurs through the release of anti-inflammatory cytokines like interleukin-10 (IL-10) and transforming growth factor-β (TGF-β), as well as various growth factors, including insulin-like growth factor 1 (IGF-1) and brain-derived neurotrophic factor (BDNF).

Nonetheless, the understanding of microglial activation has significantly evolved, and a more nuanced perspective has largely supplanted this binary model [13]. The former viewpoint is now recognized as a simplification, with the current view advocating that microglial activation is more accurately depicted as a continuum of functional states [13]. Observations in AD brains have revealed that microglia often express traits attributed to pro-inflammatory and anti-inflammatory states, indicating that their activation status in response to the disease environment is intricate, fluid, and dynamic [12,14,15,16].

Multiple molecular pathways and signaling mechanisms exist that are responsible for these shifts in the functional state of microglia (Table 1). For instance, danger-associated molecular patterns (DAMPs) and pathogen-associated molecular patterns (PAMPs) can activate the toll-like receptor (TLR) pathway, mainly TLR4, initiating a pro-inflammatory response [12,14,15,16]. TLRs are a family of pattern recognition receptors that play a crucial role in the innate immune system. They recognize molecules that are broadly shared by pathogens, as well as endogenous molecules released during cell damage or death. Upon activation, TLRs trigger signaling pathways that produce pro-inflammatory cytokines and chemokines, promoting inflammation and immune responses [12,14,15,16]. Microglial transition from the homeostatic state to a reactive profile can occur, including MGnD, DAM, and LDAM [17].

Neurodegenerative microglia (MGnD) are observed in many neurodegenerative models, including the APP-PS1 model, which mimics the pathological conditions of AD [17]. This phenotype is induced by increased expression of phagocytic genes capable of phagocytizing beta-amyloid (Aβ) plaques, apoptotic neurons, and pathological tau proteins. The TREM2 receptor is essential in inducing this state, mediating the transition of microglia from a homeostatic profile to the MGnD phenotype [17].

Lipid droplet-accumulating microglia (LDAM) are characterized by lipid droplets in aging brains due to metabolic dysfunctions and neuroinflammation [18]. In contrast, microglia without these lipid droplets show a preservation of homeostatic functions, suggesting that the LDAM phenotype is directly involved in AD progression [18]. The DAM (Disease-associated microglia) phenotype is present in neurodegenerative diseases and is characterized by functional and molecular changes, mediated especially by the TREM2 receptor [17]. The upregulation of genes associated with the immune system and lipid metabolism, such as Apoe, Clec7a, and Axl, are vital for the functional adaptation of microglia at the pathological level.

As AD progresses, microglia may gradually transition towards “dystrophic” states [12,14,15,16]. In these states, microglial processes become fragmented, and their ability to move and phagocytose—essential functions for maintaining brain health—is substantially impaired [12,14,15,16]. This could lead to an accumulation of cellular debris and amyloid-beta plaques, exacerbating inflammation and neuronal damage. Current research suggests that dystrophic microglia might contribute to the disease progression by releasing potentially harmful molecules, further amplifying neurodegeneration [12,14,15,16].

Understanding microglial activation as a spectrum of functional states offers a comprehensive and dynamic framework to elucidate their roles in the pathogenesis of AD. This model emphasizes the multifaceted nature of microglial behavior, transcending the overly simplified pro- and anti-inflammatory dichotomy and providing a more precise platform for exploring novel therapeutic strategies [12,14,15,16]. Future treatments may modulate the functional states of microglia, fostering beneficial behaviors while minimizing harmful ones. This approach could represent a promising avenue for managing and potentially modifying AD progression.

### 2.2. The Intricate Dynamics of Microglial Involvement in Amyloid-Beta Aggregation and Clearance

Microglia are crucial in building up and removing Amyloid-beta (Aβ) in AD [19]. Their reactions to Aβ, which forms plaques in the brains of AD patients, are vital to the disease’s development [19]. Neuroinflammation and microglial activation in AD have been extensively studied, revealing their critical roles in the aggregation and clearance of Aβ [12,14,15,16].

Microglia have a wide range of surface receptors essential for recognizing and binding Aβ deposits (Figure 2). The scavenger receptors SR-A, CD36, and RAGE (receptor for advanced glycation end products) are particularly significant among these receptors. These receptors initiate receptor-ligand interactions, which are the first step in phagocytosis, where microglia consume Aβ aggregates [20].

In addition to these receptors, Toll-like receptors (TLRs), primarily TLR4 and TLR2, play a crucial role in the immune response of microglia. These receptors activate signaling cascades that produce pro-inflammatory cytokines, potentially exacerbating neuroinflammation, a characteristic AD feature [20].

Scavenger receptors such as class B member 1 (SR-B1), CD36, and scavenger receptor A (SR-A) facilitate the uptake of Aβ into microglia. However, as organisms age, the effectiveness of these receptors may decline, leading to an accumulation of Aβ [20]. Moreover, RAGE contributes to the influx of Aβ across the blood-brain barrier, potentially facilitating the formation of plaques, a hallmark of AD [20].

Microglia play a vital role in the process of phagocytosis, where Aβ aggregates are internalized. This process begins with actin remodeling, followed by the formation of a phagosome, and culminates in the engulfment of Aβ [15]. Once Aβ is internalized within a microglial cell, it fuses with a lysosome to form a phagolysosome. Various lysosomal enzymes, including cathepsins, then degrade the Aβ. However, this degradation process can falter in an aged brain or AD, leading to incomplete breakdown and persistence of partially digested Aβ [15]. This scenario can trigger inflammatory responses and disrupt cellular functions, furthering disease progression [12,14,15,19].

In the early stages of AD, microglia serve as defenders, effectively recognizing and clearing Aβ. However, their role undergoes significant changes as the disease progresses. Chronic activation often occurs as AD advances, and this shift affects the microglia’s ability to continue efficiently phagocytosing and degrading Aβ. Their functional abilities decline, leading to increased Aβ plaques. This switch from an Aβ-clearing to a chronically activated state underlines the contradictory role of microglia in AD [12,14,15,19].

Emerging on this landscape are the DAM, a distinctive subset of microglia associated with neurodegenerative conditions [12,14,15,19]. They carry a unique molecular signature and contribute to both the protective and deleterious aspects of AD pathology. The roles played by DAM in AD are diverse, spanning from aiding in Aβ clearance to promoting chronic neuroinflammation and highlighting the heterogeneity within the microglial population [12,14,15,19]. DAM are marked by the downregulation of homeostatic microglial markers, such as P2ry12 and Tmem119, and the upregulation of genes typically associated with disease states, including Trem2, Apoe, and Lpl [21]. Notably, the transition from homeostatic microglia to DAM is a two-phased process initiated by signals from neurodegeneration, followed by an amplification phase dependent on the receptor TREM2 [21].

The multifaceted role of DAM in disease progression is complex. On the one hand, they play a part in containing the spread of Aβ plaques by forming a protective barrier around them, potentially limiting their growth and preventing the release of toxic substances. DAM are also implicated in the clearance of cellular debris and Aβ, though this function appears to wane as the disease progresses. Conversely, DAM might also contribute to disease progression. Although they encase plaques, their ability to completely halt the spread of Aβ might be insufficient, particularly during the advanced stages of the disease. Hence, the role of DAM in AD is paradoxical, as they might partake in both protective and potentially damaging processes. The balance between these opposing functions might shift at different disease stages, influenced by genetic background, age, and disease milieu [12,14,15,19].

The genetic landscape of AD is populated by numerous susceptibility genes that significantly contribute to the disease’s onset and progression. The *TREM2* gene is one of the most compelling, given its fundamental role in the function and behavior of microglia, which are critical players in AD pathology [22,23,24].

TREM2, or the Triggering Receptor Expressed on Myeloid Cells 2, is a transmembrane receptor protein primarily expressed in microglia within the brain. It is a molecular switch that controls several microglial functions, including their survival, proliferation, activation state, and the phagocytosis of apoptotic neurons and amyloid-beta (Aβ) plaques [22,23,24]. Moreover, TREM2 signaling aids in shifting microglia to a phenotype that promotes tissue repair and the resolution of inflammation, thereby potentially mitigating the neuroinflammatory aspects of AD [22,23,24].

However, variants of the TREM2 gene are associated with an increased risk of developing AD. Specifically, the R47H mutation in TREM2 has been identified as a strong genetic risk factor [22,23,24]. This mutation compromises the normal function of the TREM2 protein, leading to diminished phagocytic capabilities and altered microglial responses to Aβ [22,23,24].

It is also important to note that TREM2 does not function in isolation. Its activity is modulated by interactions with other proteins, such as DAP12, which is necessary for TREM2 signal transduction. Mutations in genes encoding these interacting proteins could also influence the risk of AD.

Apart from TREM2, several other genes are known to contribute to AD risk. The APOE gene, specifically the APOE ε4 allele, is perhaps the most well-known genetic risk factor for late-onset AD [25]. APOE plays a role in lipid transport and Aβ metabolism, and the ε4 variant has been associated with increased Aβ deposition [26]. Other genes, such as PSEN1, PSEN2, and APP, are implicated in early-onset familial AD, with mutations leading to increased production of toxic Aβ peptides.

The interplay between genetic and environmental or lifestyle risk factors further complicates the AD landscape. For example, cardiovascular risk factors such as hypertension, diabetes, obesity, and smoking can interact with genetic susceptibility to modulate AD risk. Furthermore, lifestyle factors such as diet, physical activity, cognitive engagement, and social interaction can also influence the disease course. These factors could alter gene expression or impact the function of proteins encoded by AD susceptibility genes, thereby modulating their effects on disease risk.

In summary, the relationship between microglia and Aβ in AD is dynamic, intricate, and highly regulated. Understanding the diverse mechanisms of microglial response to Aβ, including their recognition, binding, phagocytosis, degradation, and the factors influencing their behavior, is vital for developing effective therapeutics to manage and potentially reverse AD progression. Advancements in this field may pave the way for new approaches to addressing the puzzle of AD.

### 2.3. Microglial-Tau Interaction: An Underlying Factor Modulating Neurodegeneration in AD

Tau, a protein intricately associated with microtubules, is essential in upholding the stability and functionality of neurons. The problem arises when tau transitions into an aberrant state and aggregates, forming neurofibrillary tangles (NFTs). These NFTs are recognized as pathological hallmarks of several neurodegenerative diseases, collectively called tauopathies [27,28]. Braak staging is a method used to classify the degree of specific neurological pathologies such as AD and Parkinson’s disease (PD). The progressive development of NFTs follows the standard described by Braak, from the early stage to the advanced stage. NFT pathology accumulates at the locus coeruleus; its spread subsequently advances to the CA1 subfields and finally reaches the neocortex. The vulnerable regions mentioned above are associated with cognitive decline and neuronal degeneration [28,29,30]. Protective microglial mechanisms in response to tau aggressions are fundamental in combating the progression of neurodegenerative diseases (Figure 3). Microglia at the neuroprotective level act to remove NFTs by increasing phagocytic activity and modulating anti-inflammatory mediators that aim to reestablish neuro-homeostasis [31,32,33]. AD and frontotemporal dementia (FTD) are prominent examples. The complex and dynamic interaction between microglia and tau pathology has become a focal point of scientific exploration in this field [31,32,33].

Delving further into the molecular interplay between microglial activation and tau propagation, we find NF-κB (nuclear factor kappa-light-chain-enhancer of activated B cells) signaling at center stage [34]. The activation of tau has been found to stimulate NF-κB signaling within microglia, which in turn fosters tau propagation. Notably, the inhibition of this signaling pathway yields several therapeutic benefits: it mitigates the release of internalized pathological tau fibrils from primary microglia, ameliorates autophagy-related deficits, and reverses learning and memory impairments caused by tau [34]. The NLRP3 (NACHT, LRR, and PYD domains-containing protein 3) inflammasome also holds significant sway over tau-associated neurodegeneration, tau pathology, and tau propagation [35], further underscoring the central role of microglia in tauopathies.

Spatial patterns in disease progression have also been the subject of recent scientific inquiry. Some studies suggest that microglial activation and tau accumulation might co-localize in the human brain, following a pattern akin to the Braak stages, a system typically employed to track AD progression [36]. This spatial co-localization signifies a complex interaction among amyloid-beta, tau, and microglial activation. This tripartite interaction could dictate the pace of tau pathology spread across different Braak stages, thereby playing a significant role in cognitive decline.

Studies employing tau-seed models provide further insights. These studies suggest that amyloid pathology can potentiate bilateral tau propagation. Interestingly, the concurrent presence of amyloid, tau, and neurodegeneration pathologies—collectively referred to as the ATN (Amyloid-Tau-Neurodegeneration) triad—seems to intensify microglial activation [37,38]. However, therapeutic promise arises from findings that inhibiting the Colony-Stimulating Factor 1 Receptor (CSF1R) preferentially eliminates non-plaque-associated microglia, significantly decreasing tau pathology and neuronal atrophy [39].

Studies showing that depleting microglia results in a dramatic increase in tau seeding and spreading around plaques in TREM2 knockout mice further emphasize the crucial role of microglia in tauopathies [40]. Of particular interest is the observation that neurodegenerative microglia, known as MGnD, hyper-secrete p-tau+ extracellular vesicles during tau clearance, suggesting a potential link between amyloid plaque deposition and the worsening of tau propagation [41].

Despite an established link between Aβ-induced microgliosis and NLRP3-ASC inflammasome activation, how aggregated tau influences inflammasome activation remains largely unexplored. Early findings suggest that aggregated tau may induce ASC inflammasome activation in primary microglia after the uptake and lysosomal sorting of tau seeds. Moreover, inhibiting the NLRP3 inflammasome seems to significantly reduce tau pathology, thereby offering a promising therapeutic approach [42].

Additionally, emerging evidence indicates that mitochondrial dysfunction may influence the role of microglia in tau pathology. Microglial oxidative stress, driven by mitochondrial dysfunction, has been linked to the progression of AD. Elevated reactive oxygen species (ROS) production in microglia can exacerbate tau pathology by promoting inflammatory and cytotoxic responses [43].

The involvement of microglia in tau propagation also appears to be mediated through their phagocytic activity. Microglia can internalize tau aggregates and release them in a prion-like manner, facilitating the spread of tau pathology [44]. This mechanism underscores the dual role of microglia in both containing and propagating tau pathology [45].

Interestingly, distinct microglial activation states have been observed in response to tau pathology. Recent single-nucleus RNA sequencing studies have identified specific microglial subpopulations associated with tau pathology. These subpopulations exhibit unique transcriptional profiles, highlighting the heterogeneity of microglial responses in AD [42].

Moreover, microglial activation has been found to correlate with the spatial distribution of tau pathology. Microglial clusters are often located in regions with high tau burden, suggesting that microglial activity may influence the regional spread of tau aggregates [46,47].

In conclusion, the complex interplay among microglial activation, inflammatory signaling, and tau aggregation provides invaluable insights into the intricate pathology of AD and other tauopathies. While the activation pathways of microglia present promising therapeutic targets, it is vital to understand their dual role in disease progression and mitigation. Striking the right balance between beneficial and detrimental microglial populations and understanding the implications of selectively manipulating these populations will form the cornerstone of future research efforts.

## 3. Current Therapeutic Landscape for AD

The current therapeutic landscape for AD primarily revolves around symptom management, which aims to slow disease progression. However, these therapies have several limitations, including side effects, decreased effectiveness over time, and the inability to halt disease progression. Existing treatments are predominantly symptomatic and aim to alleviate cognitive deficits and behavioral symptoms. Despite their modest benefits, these treatments underscore the urgent need for more effective disease-modifying therapies.

### 3.1. Cholinesterase Inhibitors

Cholinesterase inhibitors are a class of drugs that work by inhibiting the action of cholinesterase, an enzyme that breaks down acetylcholine in the brain. Acetylcholine is a crucial neurotransmitter involved in various aspects of brain function, including learning, memory, and attention. In AD, cholinergic function declines, and fewer acetylcholine-producing neurons are present, contributing to the cognitive impairments seen in this condition.

Several types of cholinesterase inhibitors are used in the treatment of AD, listed below.

#### 3.1.1. Donepezil (Aricept)

Approved by the FDA in 1996, Donepezil is among the most commonly prescribed medications for AD. It is used to manage all stages of AD, from mild to severe. Donepezil is a selective acetylcholinesterase inhibitor that specifically targets acetylcholinesterase without affecting other types of cholinesterase. This selectivity makes it a potent medication with generally tolerable side effects, although some patients may still experience symptoms such as nausea, diarrhea, and sleep disturbances. Donepezil is taken orally once daily, typically in the evening, and the dosage can be adjusted based on the patient’s response and tolerance [48,49].

#### 3.1.2. Rivastigmine (Exelon)

Rivastigmine, approved by the FDA in 2000, is unique among cholinesterase inhibitors because it inhibits acetylcholinesterase and butyrylcholinesterase, another enzyme involved in the breakdown of acetylcholine. This dual action may offer a broader therapeutic benefit. Rivastigmine is utilized to manage mild to moderate AD symptoms and is also approved for treating dementia associated with Parkinson’s disease. This medication can be taken orally twice daily or applied through a transdermal patch once daily, which can help minimize gastrointestinal side effects [50].

All of these cholinesterase inhibitors can slow the worsening of symptoms for 6 to 12 months, on average, for about half of the individuals who take them. While they do not stop the progression of AD, they can temporarily improve symptoms and improve the quality of life for some people with Alzheimer’s [51]. Additionally, cholinesterase inhibitors can cause various side effects, some of which can be severe. These side effects often occur because acetylcholine involves many bodily functions, not just the brain [51]. Common side effects include nausea, vomiting, loss of appetite, increased frequency of bowel movements, and muscle cramps. These side effects are usually more severe when starting the medication or increasing the dose and may lessen over time. In some cases, cholinesterase inhibitors can also cause more severe side effects, such as a slow heart rate (bradycardia), fainting, difficulty passing urine, seizures, worsening of lung conditions in people with asthma or chronic obstructive pulmonary disease, and even potentially life-threatening allergic reactions.

In conclusion, cholinesterase inhibitors are essential in managing AD, especially in its early to moderate stages. By inhibiting the enzyme that breaks down acetylcholine, these drugs can temporarily improve or stabilize cognitive symptoms associated with Alzheimer’s. However, it is important to note that these medications only provide symptomatic relief and do not address the underlying pathology or halt the progression of Alzheimer’s. The effectiveness of these drugs tends to diminish over time as the disease progresses, and they come with several side effects [51].

### 3.2. NMDA Antagonists

Research into the use of NMDA inhibitors for AD is ongoing, and one drug, memantine, has been approved by the FDA and is in use. NMDA inhibitors, such as memantine, work by blocking NMDA (N-Methyl-D-aspartate) receptors in the brain, which are a type of glutamate receptor. Glutamate is a neurotransmitter, a chemical that nerves use to communicate. The theory behind the use of NMDA inhibitors in AD is based on the fact that excessive amounts of glutamate, which can be released in response to the injury and inflammation associated with AD, can lead to overexcitation of nerve cells and cause damage or death to these cells.

Memantine is used in moderate to severe cases of AD [52]. Studies show that memantine can help improve memory, attention, reason, language, and the ability to perform simple tasks. It can also help control the symptoms of AD, but it does not cure or stop the disease from progressing [53]. Beyond memantine, more research is needed to discover and develop other NMDA inhibitors that may be useful in treating AD [53]. Ongoing research explores various approaches, including different types of NMDA inhibitors, combination therapies, and treatments that target different pathways or mechanisms involved in the disease. Nonetheless, memantine remains the primary NMDA inhibitor for AD treatment [53].

### 3.3. Amyloid Beta-Targeting Drugs

Alzheimer’s research has been heavily centered around amyloid-beta plaques. The belief that these plaques are a significant cause of neurodegeneration led to the development of several drugs designed to reduce levels of amyloid-beta in the brain. Still, thus far, the clinical success of these treatments has been limited.

#### 3.3.1. Aducanumab (Aduhelm)

The approval of aducanumab by the FDA in June 2021 marked a significant milestone in the fight against Alzheimer’s. The drug, developed by Biogen and Eisai, is the first to target amyloid-beta plaques directly. Aducanumab is a monoclonal antibody, a type of protein designed in a lab to bind to specific substances in the body. In this case, aducanumab binds to amyloid-beta plaques and prompts the body’s immune cells (microglia) to attack and clear them [54]. The theory is that by reducing the number of amyloid-beta plaques, the drug might slow the progression of Alzheimer’s. However, the path to approval was not straightforward [55,56,57].

Aducanumab’s clinical trials yielded conflicting data, with one Phase III trial showing a slight benefit in slowing cognitive decline while the other showed no significant effect [55,56,57]. Additionally, the reduction in amyloid plaques (the drug’s “surrogate endpoint”) did not correlate with clinical benefits, leading to a split within the scientific community about the relevance of the amyloid hypothesis [55,56,57]. The FDA approval was conditional and required a post-approval study to confirm the drug’s efficacy.

#### 3.3.2. BACE Inhibitors

Another approach to tackling amyloid-beta is to inhibit its production. BACE inhibitors, such as verubecestat by Merck and lanabecestat by AstraZeneca/Eli Lilly, were designed to do just this by blocking the action of beta-secretase, an enzyme crucial in the formation of amyloid-beta peptides [58,59]. The idea was that by decreasing the amyloid-beta production, fewer plaques would form, and neurodegeneration might be slowed. Unfortunately, the results from large-scale clinical trials have been disappointing [58,59]. Not only did the BACE inhibitors fail to slow cognitive decline, but some trials also showed a worsening of cognitive scores, raising questions about potential off-target effects or the possibility that inhibiting amyloid-beta production might have detrimental consequences [58,59].

#### 3.3.3. Amyloid Vaccines

Immunotherapy, or using the body’s immune system to combat disease, has also been applied to Alzheimer’s. One such approach is to create a vaccine that stimulates an immune response against amyloid-beta. The first attempt, AN1792, developed by Elan/Wyeth [60], had to be discontinued due to serious side effects [61]. Following AN1792, efforts were made to develop vaccines that could prompt a more controlled immune response. While several such vaccines have progressed to clinical trials, results have yet to show substantial benefits in slowing cognitive decline [62].

#### 3.3.4. Gamma-Secretase Inhibitors/Modulators

Another approach to decreasing the production of amyloid-beta is to inhibit or modulate gamma-secretase, an enzyme that, like BACE, plays a key role in the production of amyloid-beta peptides [63]. However, gamma-secretase inhibitors, like semagacestat (Eli Lilly) and avagacestat (Bristol Myers Squibb), were found to have serious side effects in clinical trials, including an increased risk of skin cancer and gastrointestinal issues [64]. Consequently, they have been discontinued. However, research into gamma-secretase modulators, which aim to alter rather than completely inhibit the enzyme’s function, continues.

The repeated failure of drugs targeting amyloid-beta to deliver significant clinical benefits in AD has led to increased scrutiny of the amyloid hypothesis and a more substantial push to explore other targets. This includes tau protein, neuroinflammation, and various metabolic processes. Emerging from these experiences is an evolving view of AD as a multifaceted and complex disorder, potentially requiring a similarly multidimensional treatment approach. There is a growing interest in strategies aimed at intervening earlier in the disease process, even before symptoms become evident.

## 4. Potential Therapeutic Strategies in AD Involving Microglia

AD is a multifactorial disorder in which neuroinflammation and microglial dysfunction play pivotal roles in its pathogenesis and progression. Consequently, targeting microglia presents a promising avenue for developing novel therapeutic strategies. This section explores various approaches to modulating microglial function to mitigate AD pathology.

### 4.1. Enhancing Microglial Phagocytosis

Microglia are essential for clearing Aβ plaques and other toxic protein aggregates through phagocytosis. Enhancing this phagocytic ability can potentially reduce Aβ burden and slow disease progression.

Transcription Factor EB (TFEB) Activation: TFEB is a master regulator of lysosomal biogenesis and autophagy. TFEB, in response to stressful stimuli, is dephosphorylated and translocated to the nucleus, where it activates target genes, including the Coordinated Lysosomal Expression and Regulation (CLEAR) gene network. This is enriched with genes that positively coordinate the entire ALP (autophagy-lysosomal pathway) process. Activating TFEB has been shown to enhance lysosomal function in microglia, improving Aβ clearance and reducing neuroinflammation [65]. Pharmacological and genetic interventions have been the subject of research to activate TEFB and indirectly restore brain homeostasis [65].Small Molecule Enhancers: Compounds such as trehalose and spermidine activate autophagy pathways, aiding in the clearance of amyloid-beta (Aβ) aggregates and reducing neuroinflammation in AD. Trehalose promotes autophagosome formation and lysosomal degradation, reducing toxic protein buildup [66], while spermidine acts as an antioxidant, mitigating oxidative stress and restoring a neuroprotective microglial phenotype [67]. Together, these compounds target key pathological processes in AD, potentially slowing disease progression and enhancing neuronal resilience.Genetic Approaches: Therapeutic strategies aimed at enhancing the phagocytic efficacy of microglia and restoring homeostasis are considered innovative [68]. Overexpression of genes involved in phagocytosis, like TREM2, can enhance microglial clearance of Aβ [69]. Gene therapy techniques are being explored to upregulate such genes in microglia.

### 4.2. Modulating Microglial Activation States

Microglia exhibit a spectrum of activation states. Modulating these states can influence AD pathology.

Anti-inflammatory Agents: Drugs like minocycline can shift microglia from pro-inflammatory to anti-inflammatory, reducing neuroinflammation and potentially protecting neurons [70]. Minocycline, an antibiotic from the tetracycline group, is currently being investigated in phase II clinical trials [71].Cytokine Modulators: Targeting pro-inflammatory cytokines, specifically IL-1β and TNF-α, offers a promising strategy to mitigate microglial-mediated neuroinflammation in AD. Elevated levels of IL-1β and TNF-α have been implicated in the progression of AD, contributing to neuronal damage and cognitive decline [72]. TNF-α inhibitors like etanercept have shown promise. Preclinical studies suggest that etanercept can reduce brain inflammation, improve synaptic function, and alleviate cognitive deficits associated with AD [73]. Despite these advancements, challenges remain, including ensuring efficient delivery across the blood-brain barrier and minimizing systemic side effects. Ongoing research is focused on developing novel delivery systems to enhance the specificity and efficacy of these therapeutic approaches [74].Metabolic Modulators: Metabolic modulators, such as metformin, have demonstrated the ability to influence microglial activity, promoting an anti-inflammatory phenotype and enhancing phagocytic function. Metformin achieves these effects primarily through activating AMP-activated protein kinase (AMPK) and inhibiting the mammalian target of rapamycin (mTOR) signaling pathway. Activation of AMPK leads to the suppression of pro-inflammatory responses and the promotion of autophagy, facilitating the clearance of pathological proteins like Aβ and tau. In AD models, metformin has been shown to reduce Aβ deposits and limit tau pathology by enhancing microglial autophagy capabilities [75]. Additionally, metformin’s modulation of the AMPK/mTOR pathway contributes to the attenuation of neuroinflammation, further supporting its potential therapeutic role in neurodegenerative diseases [76].

### 4.3. Targeting Microglial Receptors

Microglial receptors play crucial roles in activation, phagocytosis, and inflammatory responses. Targeting these receptors offers a direct means of modulating microglial function.

TREM2 Agonists: Agonistic antibodies targeting TREM2 have been developed to enhance microglial responses to Aβ plaques, thereby promoting their protective functions. Preclinical studies have demonstrated that TREM2 activation can lead to increased microglial clustering around Aβ plaques, enhanced phagocytic activity, and a shift towards an anti-inflammatory phenotype, collectively contributing to reduced plaque burden and neurotoxicity [77]. However, the efficacy of TREM2 agonists appears to diminish in the advanced stages of AD. In later stages, microglia may become dysfunctional or adopt a disease-associated phenotype that is less responsive to TREM2 stimulation. Additionally, chronic exposure to high levels of Aβ and tau pathology can lead to microglial exhaustion, limiting the therapeutic potential of TREM2 activation in advanced AD [78]. Therefore, while TREM2 agonists offer a versatile approach and can be combined with other therapeutic strategies, their effectiveness may be constrained as the disease progresses. Ongoing research aims to optimize the timing and delivery of TREM2-targeted therapies to maximize their benefits, potentially incorporating them into combination treatments that address multiple aspects of AD pathology. Understanding the dynamic role of microglia and TREM2 throughout the course of AD is essential for developing effective therapeutic interventions [79].CD33 Inhibitors: CD33, a transmembrane receptor expressed on microglia, plays a significant role in AD pathology by negatively regulating microglial phagocytosis. Elevated CD33 expression in AD is associated with impaired clearance of amyloid-beta (Aβ) plaques, contributing to disease progression. Inhibiting CD33 function has emerged as a potential therapeutic strategy to enhance microglial-mediated Aβ clearance. Studies have demonstrated that reducing CD33 expression in microglia can increase their phagocytic activity, leading to decreased Aβ accumulation and plaque burden. For instance, gene therapy approaches utilizing adeno-associated virus vectors to deliver microRNA targeting CD33 have shown promising results in reducing Aβ levels and neuroinflammation in AD mouse models [80]. Additionally, research indicates that the short isoform of CD33 enhances Aβ1–42 phagocytosis in microglia, suggesting that modulation of CD33 isoform expression could be a viable therapeutic approach [81]. These findings underscore the therapeutic potential of CD33 inhibitors in modulating microglial function to mitigate AD pathology.CX3CR1 Modulators: The CX3C chemokine receptor 1 (CX3CR1) and its ligand, fractalkine (CX3CL1), play a pivotal role in modulating microglial activity within the CNS. Neurons predominantly express CX3CL1, while microglia express CX3CR1, facilitating neuron–microglia communication essential for maintaining CNS homeostasis [82]. Under physiological conditions, the CX3CL1–CX3CR1 axis maintains microglial quiescence, preventing unwarranted inflammatory responses. In pathological states, such as AD, this signaling pathway modulates microglial activation, influencing neuroinflammation and neuronal survival. Disruption of CX3CL1–CX3CR1 signaling has been associated with exacerbated microglial activation and increased neuronal damage, highlighting its neuroprotective role [82]. Therapeutic modulation of the CX3CL1–CX3CR1 axis holds promise for neurodegenerative diseases characterized by chronic neuroinflammation. Enhancing this signaling pathway can attenuate microglial activation, reduce the production of pro-inflammatory cytokines, and support neuronal survival. For instance, in models of cerebral ischemia, activation of the CX3CL1–CX3CR1 axis has been shown to suppress microglial activation and alleviate neuroinflammation, contributing to neuroprotective effects [83]. Conversely, disruption of CX3CL1–CX3CR1 signaling has been associated with exacerbated microglial activation and increased neuronal damage, highlighting the importance of this pathway in maintaining CNS homeostasis [82]. Therefore, therapeutic modulation of the CX3CL1–CX3CR1 axis holds promise for conditions characterized by excessive microglial activation and neuroinflammation, such as AD and other neurodegenerative disorders.

### 4.4. Reducing Microglial-Mediated Neuroinflammation

Chronic neuroinflammation, mainly driven by activated microglia, is a hallmark of AD. Strategies to reduce this inflammation are critical for therapeutic development.

NLRP3 Inflammasome Inhibitors: The NF-κB (nuclear factor kappa B) pathway plays a pivotal role in linking pathological stimuli—such as amyloid-beta (Aβ), tau, and ROS—to chronic inflammation by priming the NLRP3 inflammasome. This inflammasome is a critical mediator in producing pro-inflammatory cytokines like IL-1β within microglia, contributing to neuronal degeneration in AD. Inhibitors targeting the NLRP3 inflammasome have shown promise in mitigating neuroinflammation and Aβ pathology in AD models. MCC950, a selective NLRP3 inhibitor, effectively blocks canonical and non-canonical activation of the inflammasome. Studies have demonstrated that MCC950 can cross the BBB, reduce neuroinflammation and Aβ accumulation, and improve synaptic function in animal models of neurodegenerative diseases [84].Anti-inflammatory Nanoparticles: Nanoparticle-based delivery systems have emerged as a promising strategy to target microglia with anti-inflammatory agents. These nanoparticles can be engineered to cross the BBB and deliver therapeutic compounds directly to affected brain regions, thereby enhancing the bioavailability and efficacy of the treatments. Recent studies have demonstrated the potential of lipid nanoparticles (LNPs) in delivering small interfering RNA (siRNA) to microglial cells [85]. This approach effectively suppresses the expression of pro-inflammatory proteins linked to AD-related inflammation. For instance, a tailored LNP formulation was shown to reduce inflammation in human cell cultures and mouse models, indicating a potential new therapy for neuroinflammatory diseases [85].Additionally, mannose-coated nanoparticles have been developed to improve the targeting of microglia. These nanoparticles can deliver therapeutic agents that modulate microglial activity, thereby alleviating neuroinflammation associated with AD [86]. These advancements in nanoparticle technology offer innovative avenues for enhancing the delivery of anti-inflammatory therapies to the brain, potentially improving outcomes for patients with AD and other neurodegenerative disorders.MicroRNA-Based Therapies: MicroRNA-based therapies are being investigated to modulate neuroinflammation in AD by targeting specific microRNAs (miRNAs) that regulate inflammatory pathways in microglia. Certain miRNAs, such as miR-112, let-7b, and miR-140, are detectable in blood, cerebrospinal fluid (CSF), and brain tissues, suggesting their potential as biomarkers for AD. For instance, miR-let-7b has been reported to activate Toll-like receptors, influencing inflammatory responses in the brain [87]. While these miRNA-based strategies offer neuroprotective potential, they require validation through clinical trials to ensure specificity and safety [88].

### 4.5. Targeting Ion Channels in Microglia

Ion channels regulate various microglial functions, including activation, migration, and phagocytosis. Targeting these channels can modulate microglial activity to reduce AD pathology.

Kv1.3 Blockers: Kv1.3 potassium channels play a significant role in microglial activation, with their hyperactivity leading to the release of pro-inflammatory cytokines and reactive oxygen species (ROS). Inhibitors like PAP-1 have effectively reduced microglial activation and improved cognitive outcomes in AD models. For instance, PAP-1 treatment in APP/PS1 transgenic mice decreased neuroinflammation, reduced cerebral amyloid load, enhanced hippocampal neuronal plasticity, and improved behavioral deficits [89]. Additionally, studies have shown that Kv1.3 channels are highly expressed by microglia in human AD brains, suggesting that targeting these channels could be a viable therapeutic strategy [90]. These findings highlight the potential of Kv1.3 channel inhibitors as therapeutic agents in modulating microglial activity and mitigating AD pathology.P2X7 Receptor Antagonists: P2X7 receptors, ion channels activated by ATP and prevalent in central nervous system (CNS) microglia, significantly release pro-inflammatory cytokines. Antagonizing these receptors can mitigate neuroinflammation and safeguard neurons. Compounds such as JNJ-54175446 and JNJ-55308942 have demonstrated efficacy in preclinical studies by inhibiting P2X7 receptors, thereby reducing neuroinflammatory responses [91]. JNJ-54175446, a selective and brain-penetrant P2X7 receptor antagonist, has undergone clinical evaluation. In human trials, it exhibited the ability to cross the blood-brain barrier (BBB) and was well-tolerated across various doses [92]. Similarly, JNJ-55308942 has been investigated for its potential therapeutic effects. While preclinical studies have shown promise, further clinical trials are essential to confirm its efficacy and safety in humans. Ongoing research aims to validate the therapeutic potential of P2X7 receptor antagonists in neurodegenerative diseases and to develop effective treatments that minimize adverse effects.Calcium Channel Modulators: Ion channels, including those for calcium, sodium, and potassium, are integral to microglial functions such as proliferation, migration, and the secretion of inflammatory cytokines. Modulating these channels, particularly calcium channels, can influence microglial activation states and phagocytic activity, offering potential therapeutic interventions. Calcium signaling is pivotal in regulating microglial activities. Alterations in intracellular calcium levels can affect phagocytosis, cytokine release, and migration. Therapeutic strategies targeting calcium signaling pathways, such as using calcium channel blockers, show promise in alleviating microglial activation and slowing disease progression [93]. Potassium channels also play a significant role in microglial function. For instance, the Kv1.3 potassium channel is involved in microglial activation, and its inhibition has been shown to reduce microglial activation and improve cognitive outcomes in AD models. Modulating potassium channels may thus represent a viable therapeutic strategy for controlling microglial activity in neurodegenerative diseases [94].In summary, ion channels expressed by microglia are crucial in controlling their functions. Modulating these channels, especially calcium and potassium channels, offers potential therapeutic strategies for influencing microglial activity and treating neurodegenerative diseases.

### 4.6. High-Content Screening Assays for Microglial Modulators

Developing robust screening assays is essential for identifying potential drugs that can modulate microglial activity effectively.

Automated Imaging Systems: Automated imaging systems represent an essential advancement, integrating high-resolution microscopy with automated platforms capable of evaluating diverse cellular parameters, including morphology, signaling pathways, and functional responses. These systems enable efficient screening of large compound libraries, helping identify molecules that enhance microglial phagocytosis, suppress pro-inflammatory cytokines, or promote anti-inflammatory phenotypes. Time-lapse imaging incorporated into these platforms offers dynamic insights into the kinetics of drug-induced changes. For example, widely utilized platforms like Cellomics Arrayscan and ImageXpress allow detailed studies of microglial behavior in co-culture systems with neurons or astrocytes, closely mimicking in vivo conditions.Phenotypic Screening: Phenotypic screening provides another layer of analysis by assessing microglial morphological and functional characteristics. This approach offers a comprehensive view of microglial activation states, including their motility, cytokine release, and phagocytic activity, while also identifying compounds that might exert toxic effects on microglia. This process indirectly protects neuronal health by eliminating harmful candidates during the early phases of drug development. However, the complexity of microglial biology and their interactions with the surrounding microenvironment often pose challenges in data interpretation. Integrating advanced tools, such as single-cell RNA sequencing, into phenotypic assays can provide mechanistic insights and enhance the reliability of these screens.Machine Learning Integration: Incorporating machine learning algorithms into high-content screening processes further amplifies the analytical capacity. Machine learning facilitates the identification of subtle, non-linear correlations within large datasets that traditional statistical methods may overlook. These algorithms also enable predictive modeling, which forecasts the efficacy and safety profiles of new compounds based on training datasets. Furthermore, machine learning optimizes resource utilization by automating data analysis, reducing the time and labor required for manual curation. Convolutional neural networks (CNNs) are particularly valuable for image-based analysis, rapidly classifying microglial activation states and drug responses. Machine learning provides a comprehensive understanding of drug effects by integrating multimodal datasets, including imaging, transcriptomic, and proteomic data.

### 4.7. Precision Medicine Approaches Targeting Microglia

Personalized therapeutic strategies considering individual genetic and molecular profiles can optimize microglial-targeted treatments.

Genetic Profiling: Identifying patients with specific genetic variants, such as TREM2 mutations, can guide the selection of targeted therapies. TREM2 is a receptor expressed on microglia, and mutations in this gene are associated with an increased risk of AD due to loss of function, leading to impaired microglial response to amyloid plaques. Therapeutic strategies involving TREM2 agonists have shown efficacy in partially restoring microglial function, enhancing amyloid-beta clearance, and providing neuroprotection.Biomarker-Guided Therapies: Utilizing biomarkers that reflect microglial activation states or phagocytic capacity can help tailor treatments to individual needs and monitor therapeutic efficacy. Biomarkers associated with amyloid-beta, tau protein, and inflammatory responses are essential measurable indicators in the individualization of AD therapies. They facilitate early diagnosis and identification of pathological mechanisms, enabling timely and targeted therapeutic interventions [95].Combination Therapies: Integrating microglial modulators with other therapeutic agents, such as amyloid-beta or tau-targeting drugs, provides a multifaceted approach to AD treatment. For instance, combining anti-amyloid-beta antibodies with TREM2 agonists can enhance amyloid clearance while mitigating associated inflammatory responses. Additionally, compounds that improve lysosomal function may aid in degrading phosphorylated tau, simultaneously addressing multiple aspects of AD pathology [74].

### 4.8. Epigenetic Modulation of Microglia

Epigenetics is a multidimensional field of research that explores how environmental stimuli modify gene expression without modifying the DNA sequence. Epigenetic modifications influence microglial gene expression and function, and targeting these modifications offers a novel therapeutic strategy.

DNA Methylation Inhibitors: DNA methylation involves adding methyl groups to cytosine residues in DNA, typically leading to gene repression. In AD, aberrant methylation patterns can disrupt microglial homeostasis through hypermethylation that silences essential genes or hypomethylation that activates inflammatory genes, contributing to a neurotoxic environment. Therapeutic agents that modulate DNA methylation may restore healthy microglial function by correcting these epigenetic alterations [96].Histone Deacetylase (HDAC) Inhibitors: Histone acetylation and deacetylation regulate chromatin structure and gene transcription. In AD, increased HDAC activity can suppress genes vital for microglial balance, exacerbating neurotoxicity. HDAC inhibitors can alter chromatin structure, thereby modulating gene expression and influencing microglial responses to AD pathology [97].

### 4.9. Microglial Replacement Therapy

Replacing dysfunctional microglia with healthy ones derived from stem cells is an emerging therapeutic strategy.

Stem Cell-Derived Microglia: Induced pluripotent stem cells (iPSCs) can be differentiated into microglial-like cells and transplanted into the brain to replace dysfunctional microglia and restore their protective functions. This approach involves creating iPSC-derived microglia from patient cells, facilitating personalized treatments that minimize the risk of adverse immune reactions. Recent studies have demonstrated the potential of iPSC-derived microglia in modeling microglial function and dysfunction in various neurological conditions [98].Gene Editing: Techniques like CRISPR/Cas9 allow for precise editing of specific genomic regions, enabling the repression or activation of gene expression. In the context of microglial replacement therapy, CRISPR/Cas9 can be employed to correct genetic defects in stem cell-derived microglia before transplantation, enhancing their efficacy in clearing amyloid-beta (Aβ) plaques and reducing inflammation. For instance, introducing specific mutations associated with resilience to AD into human pluripotent stem cell-derived microglia has shown promise in reducing neuroinflammation and enhancing phagocytic functions [99].

### 4.10. Targeting Microglial Metabolism

Metabolic pathways regulate microglial activation and function. Modulating these pathways can influence microglial responses in AD.

Microglial energy metabolism heavily depends on glucose, with glycolysis often upreglated during pro-inflammatory activation. Targeting glucose metabolism can affect microglial polarization in several ways.

Metformin: This drug activates AMP-activated protein kinase (AMPK) and inhibits the mammalian target of rapamycin (mTOR) pathway. As a result, it promotes anti-inflammatory microglial phenotypes and enhances autophagic clearance of Aβ and tau aggregates [100]. Preclinical studies have shown reductions in Aβ burden and decreased neuroinflammation in AD models [100].2-Deoxy-D-glucose (2-DG): As a glycolysis inhibitor, 2-DG can potentially limit pro-inflammatory responses. It shifts microglia toward oxidative phosphorylation (OXPHOS)—the dominant metabolic state, which reduces cytokine production and helps preserve neuronal health [101,102]. Studies suggest that 2-DG treatment suppresses pro-inflammatory polarization and cytokine production in activated microglia [101,102].Resveratrol: This polyphenol influences glucose metabolism by activating sirtuins and AMPK, reducing inflammatory cytokine production and oxidative stress in microglia. Research indicates that resveratrol treatment decreases glucose uptake in human microglial cells, reducing pro-inflammatory activation [103].

Mitochondria are critical regulators of microglial energy production and reactive oxygen species (ROS) generation. Enhancing mitochondrial function can mitigate oxidative stress and neuroinflammation.

Coenzyme Q10 (CoQ10): As a key component of the electron transport chain, CoQ10 supplementation improves mitochondrial efficiency and reduces ROS production. In AD models, CoQ10 has been linked to decreased microglial activation and enhanced neuronal survival [104].Nicotinamide Riboside (NR): A precursor of nicotinamide adenine dinucleotide (NAD+), NR supports mitochondrial function and has been shown to modulate microglial activation states, reducing pro-inflammatory responses and enhancing phagocytic activity in neurodegenerative settings. Studies suggest that NR supplementation can alleviate microglial activation and neuroinflammation [105].

Microglial lipid metabolism is closely linked to their functional phenotypes, with lipid droplet accumulation associated with inflammatory activation. Modulating lipid metabolism can influence microglial function.

Omega-3 Fatty Acids: Eicosapentaenoic acid (EPA) and docosahexaenoic acid (DHA) exert anti-inflammatory effects by modulating lipid mediator pathways, such as reducing prostaglandin production. Research indicates that omega-3 supplementation can enhance the phagocytosis of AD-related amyloid by microglial cells [106].Fenofibrate: A peroxisome proliferator-activated receptor-alpha (PPAR-α) agonist, fenofibrate modulates lipid metabolism and has demonstrated anti-inflammatory effects on microglia, attenuating neuroinflammation and improving cognitive outcomes in preclinical AD models [107].

Amino acids and their derivatives play vital roles in maintaining cellular homeostasis and modulating oxidative stress. Modulating amino acid metabolism can influence microglial activation.

Spermidine: This polyamine enhances autophagy and reduces neuroinflammation. Spermidine supplementation in AD models has been associated with decreased oxidative stress and improved microglial function [67].N-Acetylcysteine (NAC): As a precursor to glutathione, NAC bolsters antioxidant defenses and mitigates ROS-mediated microglial activation. Studies have shown that NAC can suppress microglial inflammation and induce neuroprotective phenotypes, suggesting its potential to modulate microglial activity [108].Taurine: An amino sulfonic acid with anti-inflammatory properties, taurine reduces microglial activation and protects neurons by modulating calcium signaling and ROS production. Research has demonstrated that taurine treatment can reduce the number of activated microglia in the hippocampus and cortex, indicating its potential to mitigate neuroinflammation [109].In addition to the metabolic biomarkers mentioned above, therapeutic strategies that increase thiamine (vitamin B1) levels might also be a good option due to the antioxidant properties that help protect against damage associated with oxidative stress [110], also evidenced in dementia [111].

While this section highlights a few metabolic modulators influencing microglial function, it is important to note that the examples discussed are representative rather than exhaustive. The selected modulators illustrate key pathways and mechanisms relevant to AD pathology. Future studies may uncover additional modulators with significant therapeutic potential, further broadening our understanding of the intricate relationship between microglial metabolism and neurodegeneration.

### 4.11. Nanotechnology-Based Approaches

Nanoparticles can be engineered to deliver therapeutic agents directly to microglia, enhancing treatment specificity and reducing side effects.

Targeted Drug Delivery: Nanoparticles functionalized with ligands that specifically bind to microglial receptors can deliver drugs that modulate microglial function effectively [112,113]. The objective of this approach is to ensure that the link between the two is specific and targeted, ensuring the delivery of the therapeutic compounds to the target cells, thus increasing the safety and tolerability of the therapy.Controlled Release Systems: Nanotechnology allows for the controlled release of therapeutic agents, ensuring sustained modulation of microglial activity over extended periods [112,113]. Unlike the previous approach, it allows continuous drug release over time without the need for an immediate therapeutic response, providing a persistent and continuous action on microglia.

## 5. Precision Diagnosis and Biomarkers in AD

Accurate and early diagnosis of AD is crucial for effective intervention, management, and the potential slowing of disease progression. Precision diagnosis, which leverages advanced biomarkers, enables the identification of AD in its preclinical and prodromal stages, facilitating personalized therapeutic approaches.

### 5.1. Importance of Early Diagnosis

Timely interventions can be given before major neuronal loss and cognitive decline, increasing treatment efficacy. New blood tests can detect AD years before symptoms, allowing for early action. It also aids in patient care planning, enabling individuals to prepare for the future and make informed care decisions, improving quality of life. Early identification is vital for enrolling participants in clinical trials for disease-modifying therapies, aiding new treatment development. It also allows preventive strategies for high-risk individuals, potentially delaying symptom onset through lifestyle changes. In summary, early detection through advanced biomarkers significantly enhances treatment outcomes, patient care, and treatment research efforts.

### 5.2. Current Biomarkers

Biomarkers are measurable indicators of biological processes, pathogenic processes, or responses to therapeutic interventions. In AD, several biomarkers have been validated for diagnostic and prognostic purposes (Table 2), providing invaluable insights into the disease’s progression and distinguishing AD from other dementias. Recognizing the significance of biomarkers associated with microglial function, in addition to the so-called classic biomarkers, is gaining prominence and becoming essential as it enhances the understanding of neurodegenerative diseases and facilitates the evaluation of targeted therapies to modulate microglial activity.

#### 5.2.1. Cerebrospinal Fluid (CSF) Biomarkers

Cerebrospinal fluid (CSF) biomarkers are pivotal in the early diagnosis and monitoring of AD. Reduced levels of amyloid-beta (Aβ)42 in CSF correlate with amyloid plaque deposition in the brain, serving as a key indicator of AD pathology. Elevated levels of total tau (t-Tau) and phosphorylated tau (p-Tau) in CSF reflect neuronal injury and tau pathology, providing insight into the neurodegenerative processes underlying AD [114].

Additionally, increased CSF neurofilament light (NfL) levels indicate neuronal damage and are associated with disease progression, offering prognostic value [115]. Recent studies have also highlighted the role of microglial activation in AD. Biomarkers such as soluble TREM2 (sTREM2) in CSF are considered indicators of microglial activity, with elevated levels observed in AD patients [116]. Together, these biomarkers not only aid in the diagnosis of AD but also provide valuable information on disease progression and the underlying pathophysiological mechanisms, including neuronal damage and microglial activation.

#### 5.2.2. Imaging Biomarkers

Imaging biomarkers have significantly enhanced the diagnosis of AD by enabling the visualization of pathological changes in the living brain. These imaging techniques detect amyloid-beta (Aβ) and tau protein accumulations and provide insights into microglial activity and the brain’s overall integrity.

PET imaging utilizes radiotracers to identify specific proteins associated with AD. Radiotracers such as Pittsburgh Compound B (PiB), Florbetapir, and Flutemetamol bind to amyloid plaques, allowing their visualization and quantification within the brain. Tau PET tracers, including AV-1451 (Flortaucipir), help map tau tangles, illuminating the distribution and density of tau pathology. These imaging methods are crucial for early detection and monitoring of the progression of the disease [117].

MRI is essential for examining structural brain changes related to AD. It analyzes patterns of brain atrophy, particularly in areas like the hippocampus and entorhinal cortex, which are among the first to be affected by AD. Functional MRI (fMRI) assesses changes in brain connectivity and activity, identifying disruptions in neural networks associated with cognitive functions. Diffusion Tensor Imaging (DTI), a specific type of MRI, evaluates white matter integrity, revealing microstructural changes associated with AD [118].

These imaging biomarkers play a vital role in the AT(N) framework, which classifies AD biomarkers into three categories: amyloid deposition (A), tau pathology (T), and neurodegeneration (N). This framework improves our understanding of the sequence of pathological events in AD and assists in developing targeted therapeutic strategies [119].

In conclusion, imaging biomarkers like PET and MRI deliver essential insights into the pathological mechanisms of AD, supporting early diagnosis, monitoring disease progression, and assessing therapeutic interventions.

#### 5.2.3. Blood-Based Biomarkers

Advancements in assay technologies have enabled the detection of AD-related biomarkers in blood, providing a less invasive diagnostic option compared to CSF collection. Altered plasma Aβ42/Aβ40 ratios indicate amyloid pathology, with decreasing ratios linked to increased amyloid deposition in the brain. Elevated levels of phosphorylated tau (p-Tau) in plasma correlate with tau pathology and neurodegeneration, offering a promising avenue for non-invasive AD diagnostics. Neurofilament light (NfL) in plasma is a marker of neuronal damage and disease progression, reflecting findings from CSF studies [120]. Recent studies have shown that plasma biomarkers, such as Aβ42/Aβ40 ratios, p-Tau, and NfL, can predict cognitive decline and the transition to AD in cognitively unimpaired individuals. These findings indicate that blood-based biomarkers may be used for the early detection and monitoring of AD progression [121]. Furthermore, the development of ultrasensitive assay technologies, such as single-molecule array (Simoa) and mass spectrometry, has improved the accuracy and reliability of plasma biomarker measurements, bringing them closer to clinical application [122].

In summary, detecting AD-related biomarkers in plasma presents a promising, less invasive alternative to CSF analysis for diagnosing and monitoring AD. Ongoing research and technological advancements continue to enhance the sensitivity and specificity of these blood-based biomarkers, facilitating their integration into clinical practice.

### 5.3. Emerging Biomarkers

Research continuously uncovers novel biomarkers that improve diagnostic accuracy and offer deeper insights into AD’s underlying mechanisms. These emerging biomarkers cover various biological processes, including inflammation, synaptic dysfunction, and genetic predispositions.

#### 5.3.1. Inflammatory Biomarkers

Given the central role of microglia and neuroinflammation in AD, inflammatory markers are being explored as potential biomarkers. Elevated levels of interleukins such as IL-6 and IL-1β may reflect ongoing neuroinflammatory processes. Additionally, increased C-reactive protein (CRP) levels have been associated with a higher risk of AD, indicating systemic inflammation’s potential contribution to neurodegeneration. Monitoring these inflammatory biomarkers not only supports the association of neuroinflammation with AD but also allows the evaluation of the efficacy of clinical interventions to balance microglial function and reduce chronic inflammation [123]. Recent studies have highlighted the significance of the IL-1β/IL-6/CRP pathway in mediating inflammation and its downstream effects on disease progression, making it a prime candidate for intervention. Targeting this pathway could potentially modulate neuroinflammatory processes in AD [123]. Furthermore, elevated levels of IL-6 have been associated with various age-related diseases, suggesting that IL-6 could be a valuable and convenient marker of peripheral inflammation in older adults with comorbidities [123].

In summary, monitoring inflammatory biomarkers such as IL-6, IL-1β, and CRP provides valuable insights into the neuroinflammatory processes associated with AD. It offers potential avenues for therapeutic intervention aimed at modulating microglial function and reducing chronic inflammation.

#### 5.3.2. Synaptic Biomarkers

Markers of synaptic dysfunction provide essential insights into early neuronal damage associated with AD. Neurogranin, a postsynaptic protein linked to synaptic plasticity, is found at elevated levels in the CSF of AD patients, indicating synaptic loss [124]. Similarly, synaptotagmin, a protein crucial for the function and integrity of synaptic vesicles, represents synaptic changes when present in CSF. Elevated levels of these proteins in CSF are critical for understanding the synaptic modifications that lead to and accompany cognitive decline in AD. High concentrations of neurogranin and synaptotagmin signify increased synaptic loss, worsened by disrupted microglial activity that hampers communication between nerve cells. Recent research has shown that CSF neurogranin levels are significantly elevated in AD patients compared to healthy individuals, correlating with cognitive decline and brain atrophy [125]. This indicates that neurogranin might be a reliable biomarker for synaptic degeneration in AD. Furthermore, studies suggest that microglia, which act as the brain’s resident immune cells, directly influence synapse elimination during AD progression. Activated microglia can facilitate the phagocytosis of synaptic structures, leading to synaptic loss and contributing to cognitive impairments.

In summary, tracking CSF levels of neurogranin and synaptotagmin provides valuable insights into synaptic dysfunction in AD. These biomarkers, combined with an understanding of microglial activity, are vital for early detection and intervention strategies to preserve synaptic integrity and cognitive function in AD patients.

#### 5.3.3. Genetic Biomarkers

Genetic variants significantly influence the risk of AD and inform personalized interventions. The apolipoprotein E (APOE) ε4 allele stands out as the strongest genetic risk factor for late-onset AD, affecting lipid transport and amyloid-beta metabolism. Variants in the TREM2 gene are linked to altered microglial function and heightened AD risk, underscoring the relationship between genetics and immune responses in AD pathology. Understanding these genetic biomarkers enables risk stratification and the creation of targeted therapeutic strategies [23,126]. Recent studies have further clarified the role of APOE ε4 in AD. For instance, research shows that the presence of the APOE ε4 allele is linked to increased Aβ burden and may heighten vulnerability to progressive tau accumulation within the AD spectrum, independent of Aβ [127]. Similarly, TREM2 variants have been shown to impact microglial responses to AD pathology. Loss-of-function mutations in TREM2 impair microglial activation and clustering around amyloid plaques, potentially exacerbating disease progression [23].

In summary, genetic variants such as APOE ε4 and TREM2 play critical roles in AD risk and progression. Understanding these genetic factors is essential for developing personalized therapeutic strategies to modulate lipid metabolism, amyloid-beta processing, and microglial function in AD.

### 5.4. Microglia-Related Biomarkers

Microglial activation states and their associated markers are emerging as important biomarkers for AD. These biomarkers reflect the immune status of the brain and offer insights into disease progression and the effectiveness of therapeutic interventions. Soluble Triggering Receptor Expressed on Myeloid Cells 2 (sTREM2), derived from the proteolytic cleavage of the extracellular domain of TREM2, plays a crucial role in the interaction of microglia with the cerebral microenvironment. sTREM2 performs protective functions, such as stimulating innate immune responses, but it may also contribute to microglial dysfunction under certain conditions. Elevated sTREM2 levels in CSF have been linked to active microglial responses to amyloid-beta and tau pathology, correlating with disease progression and neurodegeneration [128].

In addition, components of the inflammasome, such as NLRP3 and ASC (apoptosis-associated speck-like protein containing a CARD), are being explored as biomarkers indicating ongoing neuroinflammatory processes in AD. NLRP3 is a pattern recognition receptor abundantly expressed in immune cells, including microglia. As a central part of the inflammasome, its activation and the formation of ASC specks promote the activation of caspase-1, leading to the release of proinflammatory cytokines like IL-1β and IL-18, which enhance the inflammatory response. The activation of the NLRP3 inflammasome is linked to amyloid-beta and phosphorylated tau deposition, contributing to microglial dysfunction and the progression of neurodegeneration [129].

Monitoring these biomarkers provides valuable insights into the neuroinflammatory landscape of AD, aiding in the development of targeted therapies designed to modulate microglial activity and slow disease progression.

## 6. Future Directions

AD poses challenges in neurodegenerative research, where microglia play a crucial role in both therapy and diagnostics. Key areas for enhancing our understanding of AD include characterizing microglial phenotypes. Single-cell RNA sequencing reveals diverse microglial activation states: homeostatic, reactive, and DAM. Future research should investigate the molecular mechanisms underlying these states to develop targeted therapies that promote protective phenotypes while suppressing harmful ones, aiming to identify new drug targets aligned with microglial functions. The metabolism of microglia influences their activation; exploring pathways such as glycolysis and oxidative phosphorylation could uncover potential therapeutic targets. Activating autophagy and lysosomal pathways using TFEB activators may improve microglial clearance of amyloid-beta and tau aggregates, which are key issues in AD. Genetic profiling, including markers like APOE ε4 and TREM2, can aid in tailoring therapies. Combination treatments that target multiple processes—incorporating microglial modulators alongside amyloid-beta or tau agents—signify progress in AD research. Reliable, non-invasive biomarkers are essential for the early detection of AD. Emerging blood-based markers, such as plasma p-Tau and NfL, have the potential to replace traditional CSF assays, emphasizing the need for validation across diverse populations and integration into multi-modal diagnostics that encompass imaging, biomarkers, and genetics. Utilizing AI to improve the interpretation of complex datasets can enhance early diagnosis and inform therapeutic decisions. Modulating microglial signaling pathways holds significant therapeutic potential; future studies should focus on enhancing phagocytosis while reducing inflammation. Targeting pathways such as the NLRP3 inflammasome may help to diminish neuroinflammation and neuronal damage. Ensuring the safety and specificity of microglial-targeted therapies is crucial, necessitating strategies to minimize off-target effects, optimize the timing of interventions, and conduct longitudinal trials to evaluate safety. Future research should promote cross-disciplinary collaboration and leverage advanced technologies. Longitudinal studies tracking microglial activation in at-risk individuals can provide insights into disease progression, while cross-species comparisons will enhance translational relevance. Multi-omics approaches integrating genomics and transcriptomics offer a comprehensive perspective on microglial biology in the AD-affected brain. Collaborations between academia, industry, and clinical centers can accelerate the discovery and validation of innovative therapeutic targets. By prioritizing these areas, we may advance Alzheimer’s diagnostics and treatments, ultimately improving patient outcomes and quality of life.

## 7. Conclusions

AD involves complex interactions among genetic, molecular, and environmental factors, with microglia playing a crucial role in its development. These immune cells have dual functions: they can offer neuroprotection but may also contribute to disease progression under specific conditions. Recent advances in microglial biology shed light on their roles in neuroinflammation, amyloid-beta clearance, and synaptic regulation, influencing new AD diagnosis and treatment strategies.

Incorporating biomarkers like sTREM2, p-Tau, and neurofilament light into AD research has transformed diagnostic and therapeutic methods. Genetic factors, including variants in APOE and TREM2, aid in developing targeted treatments. These innovations highlight the promise of personalized medicine for individuals at risk of or living with AD.

Nevertheless, challenges persist. The complexity of microglial phenotypes and the heterogeneity of AD pathology necessitate ongoing research. A multidisciplinary approach incorporating technologies such as single-cell RNA sequencing, artificial intelligence, and multi-omics is vital. Collaborative efforts among academia, industry, and clinical practice are essential to turn discoveries into meaningful patient outcomes. By advancing our understanding of microglial biology, we can reshape AD research and enhance care for those affected by this disorder.

## Figures and Tables

**Figure 1 biomedicines-13-00279-f001:**
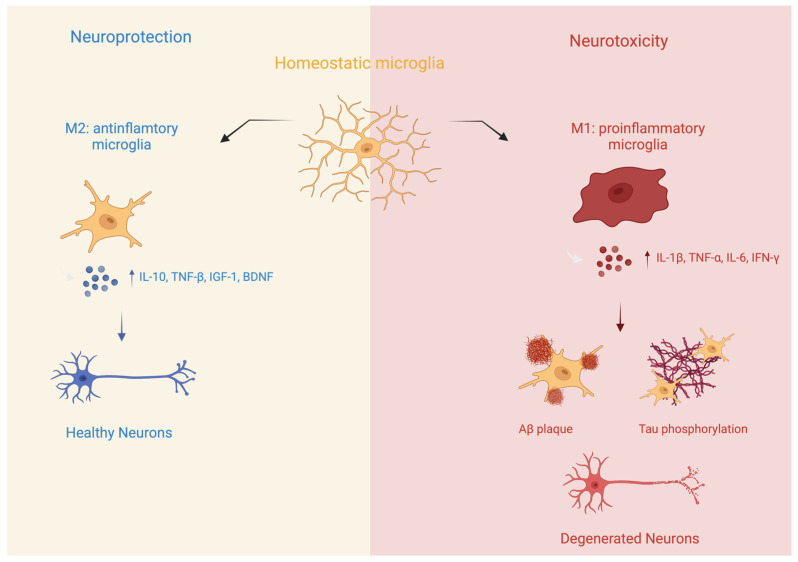
Systematic representation of the functional duality of microglia, highlighting their ability to transition between neuroprotective and neurotoxic states. In the center, the microglial balance acts as a defense mechanism in the control of the cerebral microenvironment. Dysregulation (on the right side, on a red background) exacerbates the production of inflammatory mediators, resulting in brain tissue dysfunction and neurogenetics. In contrast, microglia in the anti-inflammatory state (on the left, on a yellow background) produce neurotrophic factors that preserve neuronal integrity and functionality.

**Figure 2 biomedicines-13-00279-f002:**
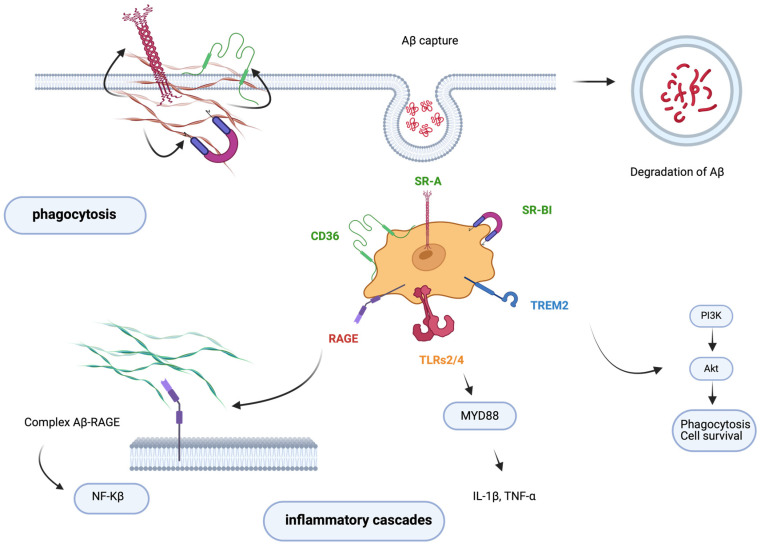
Scheme of microglial Aβ uptake, phagocytosis, and signaling employing specific receptors. The scavenger receptors represented in green, CD36, SR-A, and SR-BI favor phagocytosis and Aβ degradation intracellularly. The formation of the Aβ–RAGE complex, mediated by the receptor for advanced glycation end products (RAGE), activates the NF-kβ signaling pathway, inducing inflammatory cascades. TLRs2/4 receptors also play a role in inflammatory responses by activating the MYD88 pathway that produces pro-inflammatory cytokines. In contrast, the TREM2 receptor favors cellular survival via the PI3K/Akt pathway.

**Figure 3 biomedicines-13-00279-f003:**
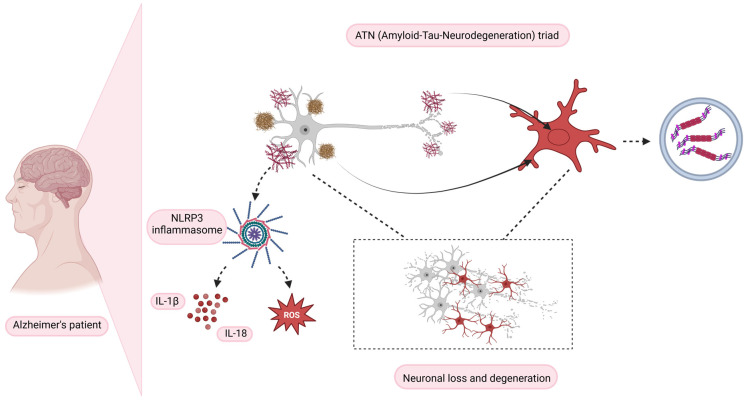
Schematic representation of microglial–tau interaction in AD. Chronic microglia activation occurs by the accumulation of beta-amyloid plaques and neurofibrillary tangles of Tau protein, resulting in neuronal loss and degeneration. This continuous inflammatory cycle promotes the activation of NLRP3, which releases inflammatory mediators and reactive oxygen species (ROS), which contribute to the cognitive decline characteristic of AD.

**Table 1 biomedicines-13-00279-t001:** MgnD, DAM, and LDAM subpopulations highlight specific features and functions in the neurodegenerative setting.

Microglial Subpopulation	Main Features	Function
MGnD (Neurodegenerative microglia)	Involved in the terminal stages of neurodegeneration	Secretion of inflammatory mediators
DAM (Disease-associated microglia)	Aids in phagocytosis of β-amyloid and tau aggregates	Regulator of the inflammatory response
LDAM (lipid droplet-accumulating microglia)	Accumulation of lipid droplets due to metabolic disorders	Intensifies oxidative stress

**Table 2 biomedicines-13-00279-t002:** Categories of biomarkers associated with AD, highlighting the biomarkers present in each category, the analytical methods used for their analysis and detection, and clinical findings.

Category	Biomarker	Analytical Method	Findings in AD
Cerebrospinal fluid (CSF) biomarkers	Aβ42	ELISA	Indicator of amyloid plaque deposition
	t-tau	ELISA, Western blotting	Intensity of neurodegeneration
	p-tau	ELISA, Immunoassays	Associated with the formation of neurofibrillary tangles
	NfL	ELISA, Western blotting	Indicate neuronal damage
Imaging biomarkers	PET (PIB, florbetapir, flutemetamol)	Positron emission tomography (PET)	Amyloid plaque detection
	Tau PET (AV-1451)	PET Imaging	Assessment of the pathological burden of tau
	MRI (fMRI, DTI)	Magnetic resonance Imaging (MRI)	Indicator of structural and functional changes
Blood-based biomarkers	Aβ42/Aβ40	Immunohistochemistry	Amyloid pathology
	p-tau	Immunohistochemistry	Tau protein pathology
	NfL	ELISA	Marker of neuronal damage
Inflammatory biomarkers	IL-6 and IL-1β	ELISA, Immunohistochemistry	Reflect active inflammatory processes
	CRP	ELISA	Marker of systemic inflammation
Synaptic biomarkers	Neurogranin	ELISA, Western blotting	Early synaptic damage
	Synaptotagmin	ELISA, Western blotting	Function and integrity of synaptic vesicles
Genetic biomarkers	APOε4	Sequencing	Influences lipid and amyloid metabolism
	TREM2	Sequencing	Changes in microglial function
Microglia-related biomarkers	sTREM2	ELISA	Indicators of microglial activation
	NLRPE and ASC	ELISA, Western blotting	Microglial inflammatory processes

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
