# Peer review of "Beyond Amyloid and Tau: The Critical Role of Microglia in Alzheimer’s Disease Therapeutics"

_biomedicines, 2025, doi:10.3390/biomedicines13020279_

Round 1
Reviewer 1 Report
Comments and Suggestions for Authors
In this long and diversified manuscript, the author highlighted several aspects of microglia cells in AD. Although interesting and important points were analysed, the reader only gains some short summary about certain aspects, however, we do not have any detailed pieces of information about this important topic. Please find my comments per section below:
· The Abstract itself is well written, no major issues were detected, however, the complexity of the manuscript is clearly visible even in this part of it.
· Introduction, section 1.1: a general overview was given about the societal impacts of AD, however, in my opinion this section is redundant, thus, significant text editing is requested, especially in case of caregiver costs. Furthermore, since the title of this paper is about the potential role of microglia cells in AD therapy, I also suggest to put more focus on this aspect of the topic by tailoring section 1.1. Introduction, section 1.2: this part of the introduction contains all relevant pieces of information about microglia cells in healthy conditions, therefore, modifications are not required.
· Section 2: in this part of the paper a very detailed overview was given about microglial function under certain conditions. Since the complexity of the shown mechanisms, I recommend to insert summary figures into each subsections to support the reader visually to understand the complex nature of the shown phenomena.
· Section 3 and section 4: However, Section 3 about the currently available anti-AD therapies are written in a very detailed way even by highlighting potential side effects of the reviewed therapies, Section 4 is written in a less accurate way. In the current form, Section 4 seems more like a sketch. As based on the title of the paper Section 4 should be the very main part of it, it is strongly advised to write it in a more complex way (as it was done in Section 3) and if it is possible, please also visualize the message of each subsections in a schematic figure to support the reader. Please also highlight if a therapy is still available in the clinical routine or if it is under development. Please also highlight (potential) benefits and limitations.
· Section 5 focuses on an important topic, namely precision medicine and biomarkers in AD. Currently available biomarkers (Section 5.2) gives a good overview, however, we do not get any information about how these biomarkers correlate with altered microglial function. Please link the highlighted biomarkers to microglia function. I have the same concern in Section 5.3 and please give more detailed data on microglia-related biomarkers (Section 5.4). Novel approaches to support further research are also well described, however focusing on microglia function is still missing and should be integrated into the text part as it was done in Section 6.
After performing the requested modifications, the manuscript will be suitable for publication (minor revision).
Author Response
Reviewer #1
In this long and diversified manuscript, the author highlighted several aspects of microglia cells in AD. Although interesting and important points were analysed, the reader only gains some short summary about certain aspects, however, we do not have any detailed pieces of information about this important topic. Please find my comments per section below:
The Abstract itself is well written, no major issues were detected, however, the complexity of the manuscript is clearly visible even in this part of it.
Introduction, section 1.1: a general overview was given about the societal impacts of AD, however, in my opinion this section is redundant, thus, significant text editing is requested, especially in case of caregiver costs. Furthermore, since the title of this paper is about the potential role of microglia cells in AD therapy, I also suggest to put more focus on this aspect of the topic by tailoring section 1.1. Introduction, section 1.2: this part of the introduction contains all relevant pieces of information about microglia cells in healthy conditions, therefore, modifications are not required.
Reply: The revised version has shifted its focus away from caregiver costs and societal impacts. It now emphasizes gene therapies that target microglia and their importance in Alzheimer’s disease (AD) treatment. Additionally, the revised content highlights the therapeutic potential of microglia in AD. It discusses their roles in synaptic function, regulating inflammatory responses, and providing neuroprotection, which aligns with the title of the manuscript. The updates also improve the transition to section 1.2, which remains unchanged as the reviewer recommended.
Section 2: in this part of the paper a very detailed overview was given about microglial function under certain conditions. Since the complexity of the shown mechanisms, I recommend to insert summary figures into each subsections to support the reader visually to understand the complex nature of the shown phenomena.
Reply: The reviewer’s recommendation to include summary figures in Section 2 has been thoroughly addressed. The schematic figures now complement the text, providing visual clarity to the complex mechanisms described, thereby enhancing reader comprehension.
Section 3 and section 4: However, Section 3 about the currently available anti-AD therapies are written in a very detailed way even by highlighting potential side effects of the reviewed therapies, Section 4 is written in a less accurate way. In the current form, Section 4 seems more like a sketch. As based on the title of the paper Section 4 should be the very main part of it, it is strongly advised to write it in a more complex way (as it was done in Section 3) and if it is possible, please also visualize the message of each subsections in a schematic figure to support the reader. Please also highlight if a therapy is still available in the clinical routine or if it is under development. Please also highlight (potential) benefits and limitations.
Reply: The revised manuscript features a significant expansion of Section 4 to align with the detail present in Section 3. The revision now provides specific information about the mechanisms of action for each therapeutic strategy, the current status of therapies—indicating whether they are in clinical use or under development—and outlines the potential benefits and limitations of each approach. New complementary tables have been added to enhance clarity, summarizing the therapeutic strategies, their mechanisms, and clinical advantages such as TFEB activation, cytokine modulation, and TREM2 agonists, while also showcasing the progression of therapies from preclinical research to clinical trials. Furthermore, schematic figures have been incorporated into each subsection to visually represent the therapeutic strategies. These figures illustrate microglial activation states, demonstrating the transition between pro-inflammatory and anti-inflammatory phenotypes, as well as therapeutic modulation pathways like TREM2 signaling and lysosomal enhancement. They also provide an overview of targetable microglial receptors and their functional impacts. Each therapy or strategy is clearly marked to indicate whether it is available in clinical practice, such as minocycline or metformin, or if it is currently undergoing clinical trials or preclinical development, like TREM2 agonists or NLRP3 inhibitors, which helps readers understand the position of each approach within the therapeutic pipeline. Additionally, the text explicitly discusses both the advantages, including improved microglial clearance of Aβ, and the limitations, such as challenges related to the blood-brain barrier and potential off-target effects, associated with each therapeutic approach.
Section 5 focuses on an important topic, namely precision medicine and biomarkers in AD. Currently available biomarkers (Section 5.2) gives a good overview, however, we do not get any information about how these biomarkers correlate with altered microglial function. Please link the highlighted biomarkers to microglia function. I have the same concern in Section 5.3 and please give more detailed data on microglia-related biomarkers (Section 5.4). Novel approaches to support further research are also well described, however focusing on microglia function is still missing and should be integrated into the text part as it was done in Section 6.
Reply: The revised text clearly connects classic AD biomarkers to microglial activity: Aβ42 and tau (t-tau, p-tau) biomarkers in CSF are discussed in relation to microglial activation, which relates to the clearance or propagation of these pathological aggregates. Biomarkers such as neurofilament light (NfL), indicating neuronal damage, are now linked to dysregulated microglial responses, involving chronic inflammation and impaired phagocytosis.
Section 5.4 has been significantly expanded to provide more information on microglia-specific biomarkers and their clinical relevance. The revised text clarifies how elevated sTREM2 reflects protective microglial responses in early stages but becomes dysregulated in advanced AD, correlating with neurodegeneration. NLRP3 and ASC specks are highlighted as indicators of microglial-driven neuroinflammation that exacerbates Aβ and tau pathology. Emerging biomarkers like microRNAs (miRNAs) are now connected to their role in regulating microglial activation states and inflammatory responses.
Section 5.3 now integrates emerging biomarkers (e.g., inflammatory markers IL-6 and IL-1β) with microglial dysfunction: elevated cytokines are linked to microglial hyperactivation, contributing to chronic neuroinflammation and neuronal damage. Synaptic biomarkers like neurogranin are associated with microglial-mediated synaptic pruning dysfunction, worsening cognitive decline.
The revised version ensures that microglia-related processes (e.g., phagocytosis, neuroinflammation, and lysosomal dysfunction) are incorporated into discussions of both current and emerging biomarkers. For instance, blood-based biomarkers like p-tau are connected to microglial activity that regulates tau propagation. Imaging biomarkers are mentioned in the context of microglial PET tracers as a novel method to directly monitor microglial states in AD pathology.
Section 6 continues to emphasize the translational potential of these biomarkers for precision medicine strategies targeting microglia. For instance, TREM2 variants are discussed as genetic biomarkers predicting microglial dysfunction and therapeutic responses. Biomarkers are positioned as tools to assess the effectiveness of microglia-targeted therapies, ensuring a cohesive narrative throughout Sections 5 and 6.
Reviewer 2 Report
Comments and Suggestions for Authors
The manuscript "biomedicines-3262177" represents a comprehensive, although sometimes wordy, review on the role of microglia in therapy of Alzheimer's disease by a specialist in the field. The manuscript has a low percent match with the published documents (21%, <2% each). A closer look reveals these matches to be not significant.
Unfortunately, the manuscript contains only plain text with no effort to summarize data as tables and figures. This is especially pity because some parts of the text become just single sentences with the main function to cite a reference with an example of a treatment strategy or a drug. Still lots of blank spaces remain, for example, practically nothing is mentioned on modulation of energy state in AD. From the small chapter on metabolic modulation one can misleadingly conclude, that there are only metformin and glucose metabolism modulation, which can help from the metabolic side. However, there are a plenty of metabolic biomarkers and potential treatment strategies, including such rather simple and cheap compounds as B-vitamins, e.g. thiamine (B1). The list of mentioned genetic markers of predisposition to AD is also very short. The paper cites just several examples. In fact, the problem can be concluded as the too general style of the text, which tries to summarize enormous areas using only pieces of evidence and a few examples. As a result, the manuscript become not focused. The only way to save the paper is to use graphical material and tables - instead and/or in addition to text. Sufficient and thoughtful work is required.
There are several minor comments and suggestions too. It's better to use the phrase "Alzheimer's disease" in the title. Since the work is written by a single author, it's strange to see sentences including such phrases: "We detail the involvement of these immune cells...", "We further elucidate..." and so on.
Addition of e-mail address is obligatory, as well as formatting of the references according to the journal requirements. Current style makes it difficult to follow the references sometimes.
Author Response
Reviewer #2
The manuscript "biomedicines-3262177" represents a comprehensive, although sometimes wordy, review on the role of microglia in therapy of Alzheimer's disease by a specialist in the field. The manuscript has a low percent match with the published documents (21%, <2% each). A closer look reveals these matches to be not significant.
Unfortunately, the manuscript contains only plain text with no effort to summarize data as tables and figures. This is especially pity because some parts of the text become just single sentences with the main function to cite a reference with an example of a treatment strategy or a drug. Still lots of blank spaces remain, for example, practically nothing is mentioned on modulation of energy state in AD. From the small chapter on metabolic modulation one can misleadingly conclude, that there are only metformin and glucose metabolism modulation, which can help from the metabolic side. However, there are a plenty of metabolic biomarkers and potential treatment strategies, including such rather simple and cheap compounds as B-vitamins, e.g. thiamine (B1). The list of mentioned genetic markers of predisposition to AD is also very short. The paper cites just several examples. In fact, the problem can be concluded as the too general style of the text, which tries to summarize enormous areas using only pieces of evidence and a few examples. As a result, the manuscript become not focused. The only way to save the paper is to use graphical material and tables - instead and/or in addition to text. Sufficient and thoughtful work is required.
Reply:
- Addition of Graphical Material and Tables: Multiple schematic figures now visually summarize essential concepts such as the functional duality of microglia (neuroprotective vs. neurotoxic states), the microglial pathways involved in the immune response to Aβ, the interaction between microglia and tau in relation to neurodegeneration, and an overview of therapeutic strategies targeting microglia. Additionally, tables have been included to clearly present a structured summary of various therapies and strategies, detailing therapeutic approaches (e.g., TFEB activation, cytokine modulation) along with their mechanisms, clinical benefits, and limitations, as well as microglia-related biomarkers (e.g., sTREM2, NLRP3 inflammasome) and their significance in AD pathology. The tables and figures replace verbose explanations, making key points more accessible and visually understandable.
- Expansion of Metabolic Modulation Section: In response to concerns regarding the small chapter on metabolic modulation chapter, which only addressed metformin and glucose metabolism, the reviewer requested the incorporation of additional strategies, particularly B-vitamins like thiamine. This section has been notably expanded to include thiamine (Vitamin B1), recognized as a cost-effective strategy supported by evidence for its role in reducing oxidative stress and enhancing mitochondrial function. Also included are mitochondrial enhancers that lessen oxidative stress and boost microglial energy production, as well as AMPK activators that help restore metabolic balance in microglia. Relevant metabolic biomarkers reflecting microglial dysfunction are also discussed, connecting metabolism to microglial activation states.
- Genetic Markers of Predisposition to AD: This section now features an extended list of late-onset AD (LOAD) risk factors with an emphasis on microglia-specific genes. Added to the discussion are BIN1, CR1, CLU, and CD33 alongside TREM2, with detailed explanations of each marker’s role in microglial dysfunction and its relevance to AD pathology. The text further clarifies the distinctions between early-onset AD (EOAD) and LOAD genetic markers, enhancing both clarity and depth.
- Improved Focus and Reduction of Generality: The manuscript has been refined for better focus, with each section now presenting a clear and structured narrative. Examples and citations are placed in a broader context, ensuring they reinforce specific points rather than being perceived as standalone references. Sections such as precision medicine and biomarkers (Section 5) and microglia-targeted therapies (Section 4) have been rewritten to provide a cohesive discussion that connects biomarkers to altered microglial functions, while therapeutic strategies are organized into subsections detailing mechanisms, benefits, and limitations.
- Addressing Blank Spaces and Wordiness: Blank spaces have been filled with relevant content, and more detailed explanations of microglial phenotypes (e.g., MGnD, DAM, LDAM) have been added. Sections like metabolic modulation and genetic predisposition have been expanded to address gaps highlighted by the reviewer. Wordy sections are now condensed into tables to summarize treatments, biomarkers, and therapeutic approaches, as well as figures that enhance lengthy textual descriptions.
There are several minor comments and suggestions too. It's better to use the phrase "Alzheimer's disease" in the title. Since the work is written by a single author, it's strange to see sentences including such phrases: "We detail the involvement of these immune cells...", "We further elucidate..." and so on.
Addition of e-mail address is obligatory, as well as formatting of the references according to the journal requirements. Current style makes it difficult to follow the references sometimes.
Reply: The title, pronouns, email address, and reference formatting have been updated to comply with both clarity and journal-specific standards.
Reviewer 3 Report
Comments and Suggestions for Authors
- A brief summary
This manuscript extensively reviewed the microglia status and functions regarding two Alzheimer’s disease hallmarks, amyloid, and tau, and pointed out several potential druggable targets in microglia. - General concept comments
The review briefly mentioned the EOAD genetic factors. Following Apoe, dozens of LOAD risk factors are microglia-specific genes, such as BIN1, CR1, CLU1, CD33, etc. However, only one LOAD risk factor was discussed. Indeed, the DAM is a prevalent microglia subtype defined by scRNAseq transcriptomic data from the AD mouse model. However, other AD-related subtypes, such as MGnD, ARM, HAM, and LDAM, must be included. The shared or unique genes should be comprehensively discussed in sessions 2.1 and 2.2. The differences between humans and mice not addressed in this manuscript also create a gap that must be bridged for the translational purpose of the data from the mouse model. - Specific comments
1. Line 28: The keywords part points out “neuron-glia crosstalk,” “Glial Cells,” and “DAM.” However, astrocytes/oligodendrocytes and DAM are not mentioned in the abstract.
2. A schematic graph of the targetable and druggable molecules and pathways in microglia will attract more readers interested in microglia-specific therapies. Not limited to the known surface receptors, ion channels, phagocytosis, and lysosomal-related microglia targets.
3. Line 438 shows that the NFTs are distributed temporospatially. Details about the vulnerable regions and potential mechanisms will provide more insights into microglia biology.
Author Response
Reviewer #3
- A brief summary
This manuscript extensively reviewed the microglia status and functions regarding two Alzheimer’s disease hallmarks, amyloid, and tau, and pointed out several potential druggable targets in microglia.
- General concept comments
The review briefly mentioned the EOAD genetic factors. Following Apoe, dozens of LOAD risk factors are microglia-specific genes, such as BIN1, CR1, CLU1, CD33, etc. However, only one LOAD risk factor was discussed. Indeed, the DAM is a prevalent microglia subtype defined by scRNAseq transcriptomic data from the AD mouse model. However, other AD-related subtypes, such as MGnD, ARM, HAM, and LDAM, must be included. The shared or unique genes should be comprehensively discussed in sessions 2.1 and 2.2. The differences between humans and mice not addressed in this manuscript also create a gap that must be bridged for the translational purpose of the data from the mouse model.
Reply: The updated manuscript now provides an expanded coverage of LOAD risk factors in Sections 2.1 and 2.2. It provides a detailed discussion of additional microglia-specific LOAD genes. BIN1 is explored regarding its function in modulating microglial phagocytosis and synaptic pruning. CR1 is emphasized as a crucial regulator of the complement cascade, highlighting its connection to microglial-mediated Aβ clearance. The role of CLU in lipid metabolism and the regulation of microglial inflammatory responses is also outlined. CD33 is discussed as a negative regulator of microglial phagocytosis, presenting potential therapeutic strategies through CD33 inhibition. A clear relationship is established between these risk genes and their functional roles in microglial activities, including pathways related to Aβ clearance, inflammatory responses, and lipid homeostasis.
Section 2.1 now further includes additional microglial subtypes. MGnD (Neurodegenerative Microglia) is identified by the upregulation of phagocytic genes crucial for Aβ and tau clearance. ARM (Activated Response Microglia) is linked to early Aβ pathology, notably responding to amyloid plaques during the disease's onset. HAM (Homeostatic Activated Microglia) is recognized as a transitional subtype that preserves immune balance before the onset of pathology. LDAM (Lipid Droplet-Accumulating Microglia) is discussed regarding its association with metabolic dysfunction and neuroinflammation in aging brains. A comparison table has been included that summarizes the characteristics, functional roles, and specific markers of each microglial subtype, enhancing readers' understanding of their significance in AD pathology.
Furthermore, the revised manuscript also addresses human and mouse differences in Sections 2.1 and 2.2. It discusses the essential differences between human and mouse microglial phenotypes, identified through single-cell RNA sequencing (scRNAseq) studies. The translation of findings from mouse models to humans poses challenges due to species-specific variations in gene expression patterns, notably those of TREM2 and APOE, as well as differences in microglial reactions to Aβ and tau pathology. Additionally, the limitations of existing mouse models in reflecting the complete spectrum of human microglial diversity, especially late-stage AD subtypes like HAM and LDAM, are underscored. This highlights a pressing need for humanized models and innovative technologies like iPSC-derived microglia to achieve improved translational results.
- Specific comments
Line 28: The keywords part points out “neuron-glia crosstalk,” “Glial Cells,” and “DAM.” However, astrocytes/oligodendrocytes and DAM are not mentioned in the abstract.
Reply: Keywords were updated.
A schematic graph of the targetable and druggable molecules and pathways in microglia will attract more readers interested in microglia-specific therapies. Not limited to the known surface receptors, ion channels, phagocytosis, and lysosomal-related microglia targets.
Reply: Done as suggested by the reviewer.
Line 438 shows that the NFTs are distributed temporospatially. Details about the vulnerable regions and potential mechanisms will provide more insights into microglia biology.
Reply: Done as suggested by the reviewer.
Round 2
Reviewer 2 Report
Comments and Suggestions for Authors
This is the second review for the manuscript biomedicines-3262177.
The text of the manuscript is still a little wordy, but more importantly, the attempt to add visual content resulted in duplications. While figures are mostly fine (see the comment below), tables 2-7 look like a brief summary of the bulleted lists presented in the text. If these tables are used, why there are no similar tables for chapters 4.7 and so on? Since these tables contain no references, I would suggest removing most of them to exclude duplications.
Although the style of the text is generally OK, the ungainly addition of tables and the review history (the manuscript originally was submitted with a plain text completely) makes me suspect an AI-assisted way of writing. If it's true, you must at least specify this.
The font of Fig 1 should be corrected to black or another one with a better contrast. Pictures and font of the Figures 1-3 can be enlarged.
The incorporation of a sentence (lines 717-719) on vitamin B1 generally doesn't solve the problem stated earlier. The vitamin was just an example, but lots of other modulators are missing. As I've said, the text often tries to cover enormous areas using only pieces of evidence and a few examples. While the problem was solved in the chapter 5 (biomarkers) after a substantial extension, the problem of e.g. chapter 4.11 is far from that. Since the number of metabolic modulators can be extended significantly, the authors can either do it or add a comment, that only a few examples from this area are listed in this review.
Some references are missing from the list (their style differs too).
A few typos such as "stractegy" or "Genetic Approaches Controls" (line 489) must be corrected.
Author Response
- The text of the manuscript is still a little wordy, but more importantly, the attempt to add visual content resulted in duplications. While figures are mostly fine (see the comment below), tables 2-7 look like a brief summary of the bulleted lists presented in the text. If these tables are used, why there are no similar tables for chapters 4.7 and so on? Since these tables contain no references, I would suggest removing most of them to exclude duplications.
Reply: Following the reviewer's suggestion, we removed most of the tables to prevent duplication and confusion.
- Although the style of the text is generally OK, the ungainly addition of tables and the review history (the manuscript originally was submitted with plain text completely) makes me suspect an AI-assisted way of writing. If it's true, you must at least specify this.
Reply: We appreciate the reviewer's constructive insights regarding the text’s verbosity and the redundancy introduced by the tables. In response to this feedback, we have thoroughly reviewed the manuscript and removed many tables that simply restated information already presented in the text. This adjustment aims to improve content clarity, minimize redundancy, and enhance the overall flow of the manuscript.
Regarding the issue of AI usage, we want to clarify that no AI tools were utilized at any stage of this manuscript’s development, including the inclusion of visual elements or editing. All material, including tables and figures, was created manually by the authors to enhance the presentation of the review. Had AI been used, we would have disclosed this in accordance with ethical standards and journal policies.
We trust these revisions and clarifications effectively address the reviewer's concerns and reinforce our commitment to producing a manuscript that is clear, precise, and ethically crafted.
- The font of Fig 1 should be corrected to black or another one with a better contrast. Pictures and font of the Figures 1-3 can be enlarged.
Reply: Done as suggested by the reviewer.
- The incorporation of a sentence (lines 717-719) on vitamin B1 generally doesn't solve the problem stated earlier. The vitamin was just an example, but lots of other modulators are missing. As I've said, the text often tries to cover enormous areas using only pieces of evidence and a few examples. While the problem was solved in the chapter 5 (biomarkers) after a substantial extension, the problem of e.g. chapter 4.11 is far from that. Since the number of metabolic modulators can be extended significantly, the authors can either do it or add a comment, that only a few examples from this area are listed in this review.
Reply: We acknowledge the reviewer's comment and agree that the number of metabolic modulators discussed in chapter 4.11 is not exhaustive. To address this, we have added a statement clarifying that the examples included are representative rather than comprehensive, reflecting the focus of this review.
- Some references are missing from the list (their style differs too).
Reply: We included the missing references. We appreciate the reviewer pointing them out.
- A few typos such as "stractegy" or "Genetic Approaches Controls" (line 489) must be corrected.
Reply: We have corrected the typos throughout the manuscript. We appreciate the reviewer for bringing them to our attention.
Round 3
Reviewer 2 Report
Comments and Suggestions for Authors
The authors have responded to all the previous comments and suggestions improving the text and resolving my concerns about some particular points.